# Structural basis for the inhibition of translation through eIF2α phosphorylation

Yuliya Gordiyenko[1,3], José Luis Llácer [1,2,3] & V. Ramakrishnan [1]

One of the responses to stress by eukaryotic cells is the down-regulation of protein synthesis by phosphorylation of translation initiation factor eIF2. Phosphorylation results in low availability of the eIF2 ternary complex (eIF2-GTP-tRNAi) by affecting the interaction of eIF2 with its GTP-GDP exchange factor eIF2B. We have determined the cryo-EM structure of yeast eIF2B in complex with phosphorylated eIF2 at an overall resolution of 4.2 Å. Two eIF2 molecules bind opposite sides of an eIF2B hetero-decamer through eIF2α-D1, which contains the phosphorylated Ser51. eIF2α-D1 is mainly inserted between the N-terminal helix bundle domains of δ and α subunits of eIF2B. Phosphorylation of Ser51 enhances binding to eIF2B through direct interactions of phosphate groups with residues in eIF2Bα and indirectly by inducing contacts of eIF2α helix 58–63 with eIF2Bδ leading to a competition with Met-tRNAi.

[1] MRC Laboratory of Molecular Biology, Francis Crick Avenue, Cambridge CB2 0QH, UK. [2] Instituto de Biomedicina de Valencia del Consejo Superior de Investigaciones Científicas and CIBERER-ISCIII, Valencia 46010, Spain. [3] These authors contributed equally: Yuliya Gordiyenko, José Luis Llácer. Correspondence and requests for materials should be addressed to J.L.L. (email: jllacer@ibv.csic.es)

n eukaryotes, initiation of protein synthesis is tightly regulated by a number of translation initiation factors (eIFs) including the GTPase eIF2. During initiation, the GTP-bound eIF2 forms a ternary complex (TC) with Met-tRNAi[Met], and together with other initiation factors binds the 40S ribosomal subunit, forming the 43S pre-initiation complex (PIC). Another initiation factor is the GTPase activating protein (GAP) eIF5, which promotes GTP hydrolysis by eIF2[1–4], and helps to locate the AUG start codon at the P site during scanning along mRNA[5]. After the PIC recognition of the initiation codon, inorganic phosphate is released[3], and the GDP-bound eIF2 dissociates from the 40S along with most other initiation factors. The subsequent binding of eIF5B promotes joining of the 60S and the start of the protein synthesis. For multiple rounds of initiation to occur, the GDP on eIF2 has to be exchanged for GTP. This reaction is catalysed by a guanine nucleotide exchange factor (GEF) eIF2B.

eIF2 comprises three subunits, eIF2α, eIF2β and eIF2γ. Of these, eIF2γ has the catalytic site for GTPase activity and recognises and binds the acylated acceptor arm of the Met-tRNAi[Met6,7]. eIF2β forms part of the nucleotide-binding pocket in eukaryotes[7], whereas eIF2α is inserted in the E site of the 40S subunit during translation initiation while being bound to Met-tRNAi[Met7–9], and also has a regulatory function[10,11]. In response to various stress conditions eukaryotic cells regulate protein synthesis by phosphorylation of serine 51 (52 sequence numbering) on the eIF2α, thereby converting eIF2 from a substrate to an inhibitor of its GEF, eIF2B[12,13]. This highly conserved mechanism, called integrated stress response (ISR) in mammals or general amino acid control (GAAC) in yeast, shuts down bulk protein synthesis[10,14] due to the low availability of the TC, and redirects cell resources to adaptive and survival pathways[15–18]. Deregulation of eIF2B function in humans leads to hypomyelination and neurodegenerative disorders[19,20].

The mechanism of nucleotide exchange by eIF2B and its inhibition by eIF2α phosphorylation has been a matter of considerable debate[12,21–27]. The regulatory subunits α, β, δ are homologous with a similar fold and form the hexameric core of eIF2B, while the catalytic subunits γ and ε assemble into heterodimers and bind peripherally on two opposite sides of the regulatory hexamer as shown in the X-ray structure of *S. pombe* eIF2B[28] and cryoEM structures of human eIF2B[26,27]. eIF2B γ and ε are homologous to each other and have two domains in common—a pyrophosphorylase-like domain (PLD) and a left-handed β helix (LβH) domain[29]. eIF2Bε in addition has a C-terminal HEAT domain extension[30]—ε-cat, which itself possesses catalytic activity[31]. This structural complexity makes it more difficult to understand the mechanism of action and regulation of eIF2B.

The interactions of eIF2 with eIF2B have been extensively investigated biochemically and genetically by mutagenesis of both factors[32–37]. In addition, the thermodynamics of eIF2-GDP recycling to the TC has also been studied[24]. Nevertheless, in the absence of a structure of the eIF2B–eIF2 complex, details of the mechanism of nucleotide exchange and its inhibition by eIF2α phosphorylation remain unclear.

Here we have determined a cryoEM structure of eIF2B in complex with the GDP-bound form of eIF2 phosphorylated at Ser51 on the α subunit, which sheds light on the molecular interactions between the two molecules and provides a basis for understanding the regulation of translation by eIF2α phosphorylation.

## Results

**An overall structure of eIF2B–eIF2(αP) complex.** Two datasets of eIF2B–eIF2(αP) complex were acquired, one in linear mode and another in counting mode (see Methods for details). The structure of eIF2B–eIF2(αP) complex was determined to an overall resolution of 4.2 Å at best using the counting mode dataset only (map 1, Supplementary Fig. 1). This structure was obtained by applying a twofold C2 symmetry during EM data processing, resulting in maximum resolution for the most homogeneous parts of the model but an averaged position for the eIF2 molecules, which showed a high degree of conformational heterogeneity at the periphery of the complex. To improve resolution in this region, we combined particles from both datasets and carried out 3D EM data classification applying a twofold C2 symmetry and using masks around eIF2 molecules in the complex. This classification resulted in map 2 (Supplementary Fig. 1) with slightly lower overall resolution 4.3 Å, however, the local resolution for eIF2γ and eIF2α-D3 was better compared to map 1. The map obtained using a linear mode dataset only did not yield a high overall resolution (5.7 Å, Supplementary Fig. 1). To further account for the different conformations of eIF2 in the complex, we also carried out focused EM data classifications using combined dataset without applying any internal symmetry and we obtained another four density maps (maps A to D in Supplementary Fig. 1, see Methods for details), however, at lower overall resolution.

The structure consists of two eIF2 molecules bound to opposite sides of the eIF2B hetero-decamer (Fig. 1a, b). Each eIF2 molecule has two spatially separated interactions with the eIF2B hetero-decamer—one through eIF2α-D1 inserted in the pocket between eIF2B α and δ subunits and another contact of eIF2γ with the catalytic eIF2B subunits. As judged by the relatively high local resolution (Supplementary Fig. 2), which reflects low local flexibility and mobility, the strongest contact consists of eIF2α domain D1 inserted between the N-terminal helix bundle domains of α and δ regulatory subunits of eIF2B. In our complex this interaction is possibly further stabilised by phosphorylation of eIF2α, which was shown previously to enhance binding to eIF2B regulatory subcomplex[12,38]. Another contact is formed by eIF2γ and eIF2β interacting with the catalytic eIF2B subunits γ and ε (Fig. 1b). This area of contact has lower local resolution, suggesting that the region has conformational heterogeneity and the interaction is very dynamic.

In a low-resolution filtered map 2 contoured at lower threshold, we could see weak densities around eIF2γ, which cannot be attributed to this subunit (Supplementary Fig. 1, blue and red masks). Masked classification[39] around these densities and eIF2γ allowed us to separate different conformations that eIF2 γ and β adopt in the four different maps obtained (Supplementary Fig. 1, maps A–D). In two of these maps (B and D), we also observed in proximity to eIF2γ additional unknown low-resolution densities that could not be attributed to any region of eIF2 (see Supplementary Fig. 1, extra density in map D).

**Interaction of the phosphorylated Ser51 on eIF2 with eIF2B.** The phosphorylated Ser51 is part of the domain eIF2α-D1, and the structure provides a rationale for why phosphorylation of this residue should inhibit eIF2B function. The domain is inserted between the N-terminal helix bundle domains of δ and α subunits of one set of eIF2B subunits (Fig. 2a, b) rather than binding the central cleft of eIF2B as proposed in a previous model[28]. Interestingly, the crosslinks of eIF2α to eIF2B α and δ obtained for the model[28] are in perfect agreement with the binding of eIF2α-D1 in our structure (Fig. 2b), whereas the crosslinks to eIF2β cannot be explained in the context of our structure. Instead, in agreement with previously identified mutations I118T and S119P in eIF2Bβ that were shown to reduce the effect of eIF2α phosphorylation[40], the loop 113–120 of eIF2Bβ (coloured brown), from what could be considered another set of eIF2B subunits, participate in the contact with eIF2α-D1 (Fig. 2b).

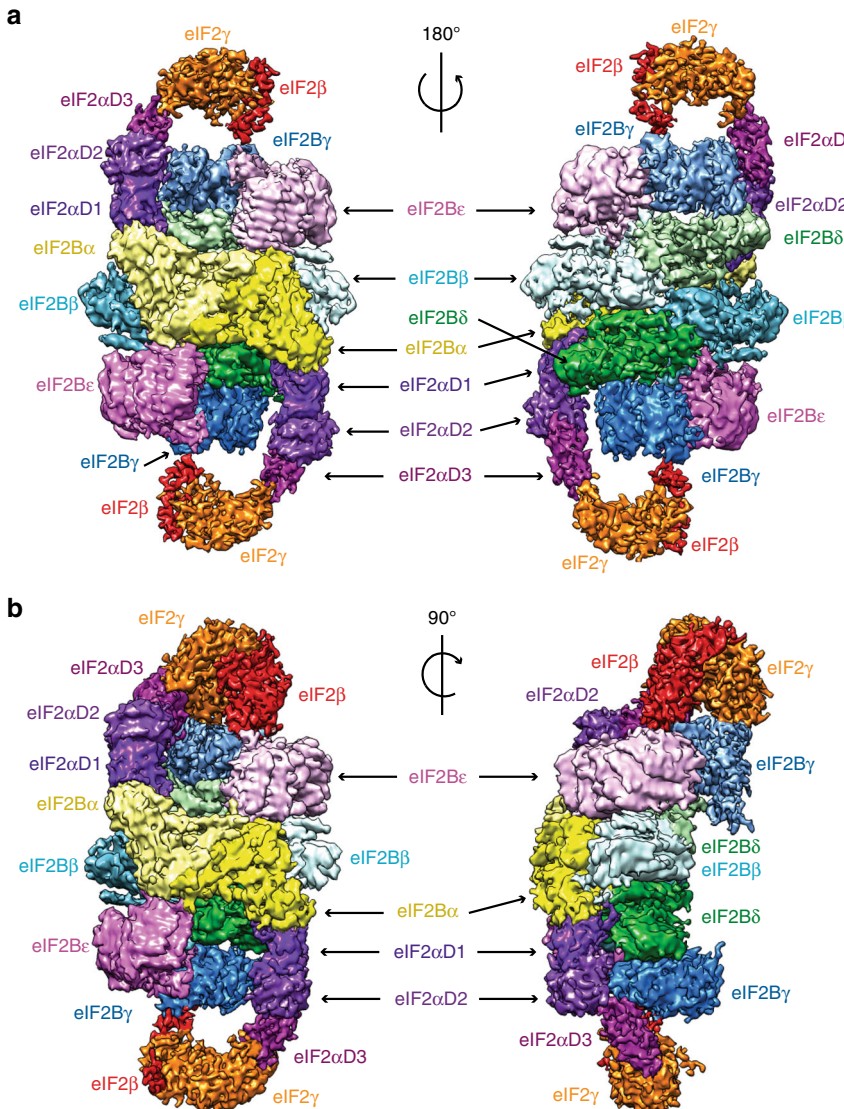

**Fig. 1** Overview of cCryoEM structure of eIF2B–eIF2(αP) complex. **a** Two views of the overall cryoEM map 2 of eIF2B–eIF2(αP) complex at 4.3 Å resolution with different subunits of the complex colour-coded. **b** Two views of the cryoEM map A of eIF2B–eIF2(αP) complex at 4.6 Å containing clear density for eIF2β subunit on one side of the complex at the top

When compared to the crystal structure of *S. pombe* eIF2B alone[28] or cryoEM structures of human ISRIB bound eIF2B[26,27] (Supplementary Fig. 3a), the binding of eIF2α-D1 in our complex leads to a closure of eIF2B δ and α helix bundle NTD domains around it (Supplementary Fig. 3b). Closure of the domains also leads to a visible displacement of eIF2Bγ PLD about 5–6 Å outwards (Supplementary Fig. 3c), making the eIF2B heterodecamer in the complex with eIF2(αP) elongated by ~10–12 Å compared to an apo form[28] or ISRIB bound human eIF2B[26,27] (Supplementary Fig. 3a). The most extensive interaction surface area (844 Å[2]) is between the eIF2α-D1 and eIF2Bα subunits, which would explain why eIF2B mutants lacking an α subunit are not sensitive to eIF2α phosphorylation, as the major part of the binding surface with eIF2α-D1 would be lost.

The density in eIF2α-D1 leading to and including the phosphorylated Ser51 is visible (Fig. 2b and Supplementary Fig. 4), however, the arginine-rich loop following this serine seems to be only partially ordered. At this resolution, we cannot establish with complete confidence the interaction partners of Ser51-P because the densities for the side chains around the residue are not absolutely clear. However, the closest residues to

the phosphate on Ser51 appear to be eIF2Bα H82 and Y304 and R75 slightly further away (Fig. 2b). Furthermore, in this position the phosphate may affect the conformation of the short α-helix 58–63 after the Arg-rich loop that in turn makes contacts with the eIF2Bδ NTD in our structure (Fig. 2b). eIF2Bδ residues E377 and L381 are likely to be involved in this interaction as mutations E377K and L381Q were shown to overcome the effect of Ser51 phosphorylation[33], suggesting that described mutations would disrupt or weaken this interaction. Indeed, mutation of the residue analogous to E377 in *S. pombe* (D248K) abrogated strong interaction of eIF2(α)P with eIF2B and alleviated inhibition of nucleotide exchange[28].

eIF2α phosphorylation is known to increase its binding affinity to eIF2B[38], and our structure suggests that this is due to a combination of direct contact of Ser51-P with residues in eIF2Bα (H82 and/or Y304 and/or a long electrostatic interaction with R75) as well as tighter induced interaction of the 58–63 α-helix with eIF2Bδ. Interestingly the same helix 58–63 contacts Met-tRNA$_i^{Met}$ in the TC structure[7,9,41], although in the TC, this helix adopts a slightly different conformation (Fig. 2c). This suggests that initiator tRNA and eIF2B compete for the same binding site

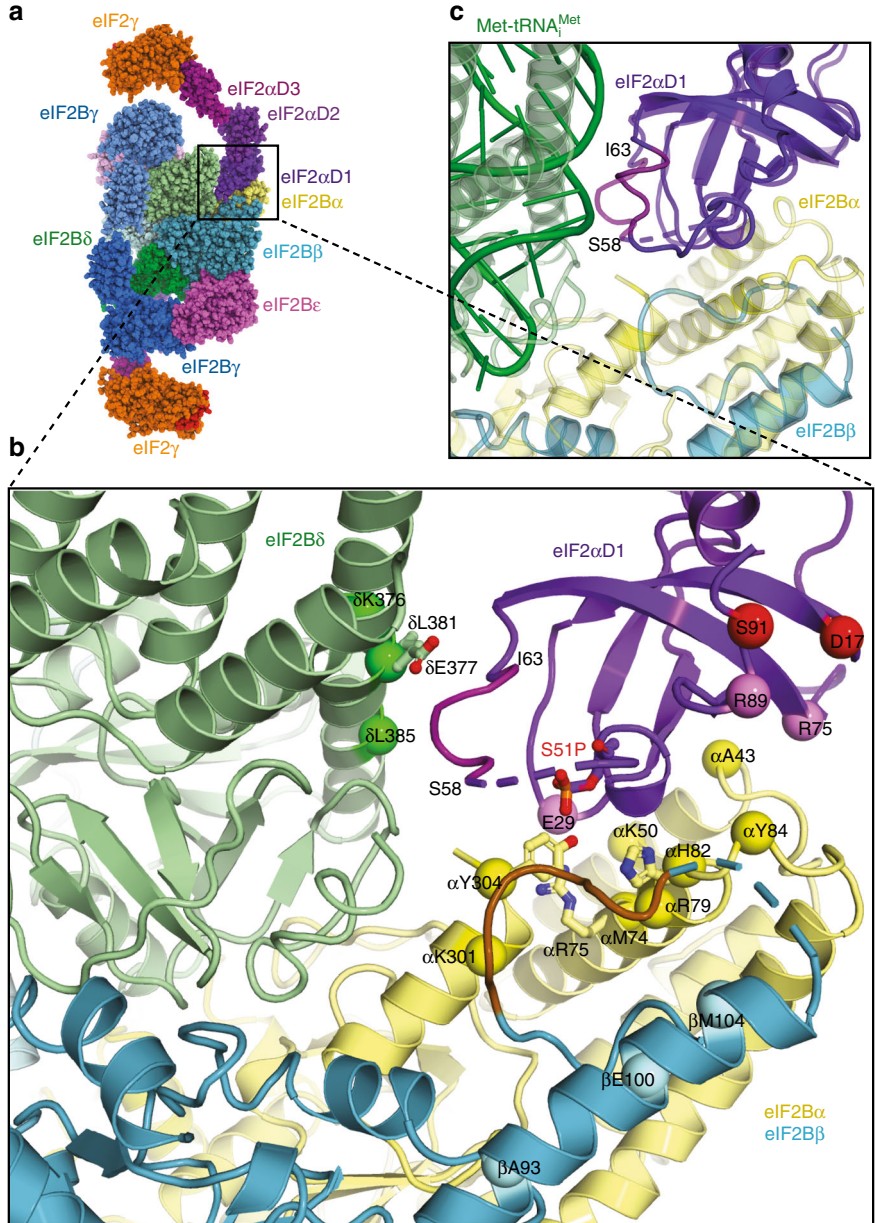

**Fig. 2** Contacts of eIF2α with the regulatory eIF2B subunits. **a** Model of eIF2B–eIF2(αP) complex fitted in maps 1 and 2. **b** Contacts of eIF2α-D1 with α, β and δ regulatory subunits of eIF2B. Possible residues in contacts with eIF2α Ser51 phosphate (red sticks) are H82, Y304 and R75 in eIF2Bα (shown in yellow sticks). eIF2Bδ E377 (green sticks) in contact with the 56–63 helix (magenta) of eIF2α affected by the phosphate is also shown. eIF2Bδ E377K overcomes the effect of Ser51 phosphorylation and eIF2Bβ I118T and S119P (in brown loop) reduce the effect of phosphorylation. Also shown are residues in *S. cerevisiae* eIF2B regulatory subunits α, β and δ corresponding to *S. pombe* residues which cross-linked to eIF2α and residues in eIF2α which cross-linked to eIF2Bα (pink spheres) and to eIF2Bβ (red spheres)[28]. **c** Superposition of eIF2α-D1 in eIF2B–eIF2(αP) complex and in the TC (PDB 3JAP), showing that the same helix 58–63 (coloured magenta) in eIF2α-D1 interacts with both eIF2Bδ and Met-tRNAᵢᴹᵉᵗ in a different conformation, suggesting direct competition for eIF2α-D1

on eIF2α, and the altered conformation of the helix upon Ser51 phosphorylation may inhibit the binding of initiator tRNA and displacement and dissociation of eIF2B.

**eIF2 γ and β interactions with catalytic eIF2B subunits.** Although eIF2α-D1 containing the phosphorylated Ser51 is relatively constrained through its interaction with eIF2B, the domains eIF2 γ and β in the proximity of the catalytic portion of eIF2B have relatively high conformational heterogeneity presumably arising from high mobility (Figs. 1b and 3c, d) and do not adopt the same conformation in two eIF2 molecules bound on either side of eIF2B (Fig. 3a). Because of this heterogeneity,

which resulted in lower resolution, we cannot be sure whether the GDP that was present in our preparations has been displaced from eIF2γ.

To separate the different conformations adopted by eIF2 γ and β, we have applied two masked classifications (Supplementary Fig. 1 and Methods). After the first masked classification, we obtained three maps A-C (Supplementary Fig. 1) with conformations of eIF2γ tilted towards the PLD domains of eIF2Bγ subunit and distinct extended conformations of eIF2β (Fig. 3a, b, f). The tilted conformation of eIF2γ is stabilised by the contacts of eIF2β with the PLD domains of eIF2B γ and ε subunits and the contact of eIF2γ domain III (eIF2γ-D3) with the eIF2Bγ PLD domain

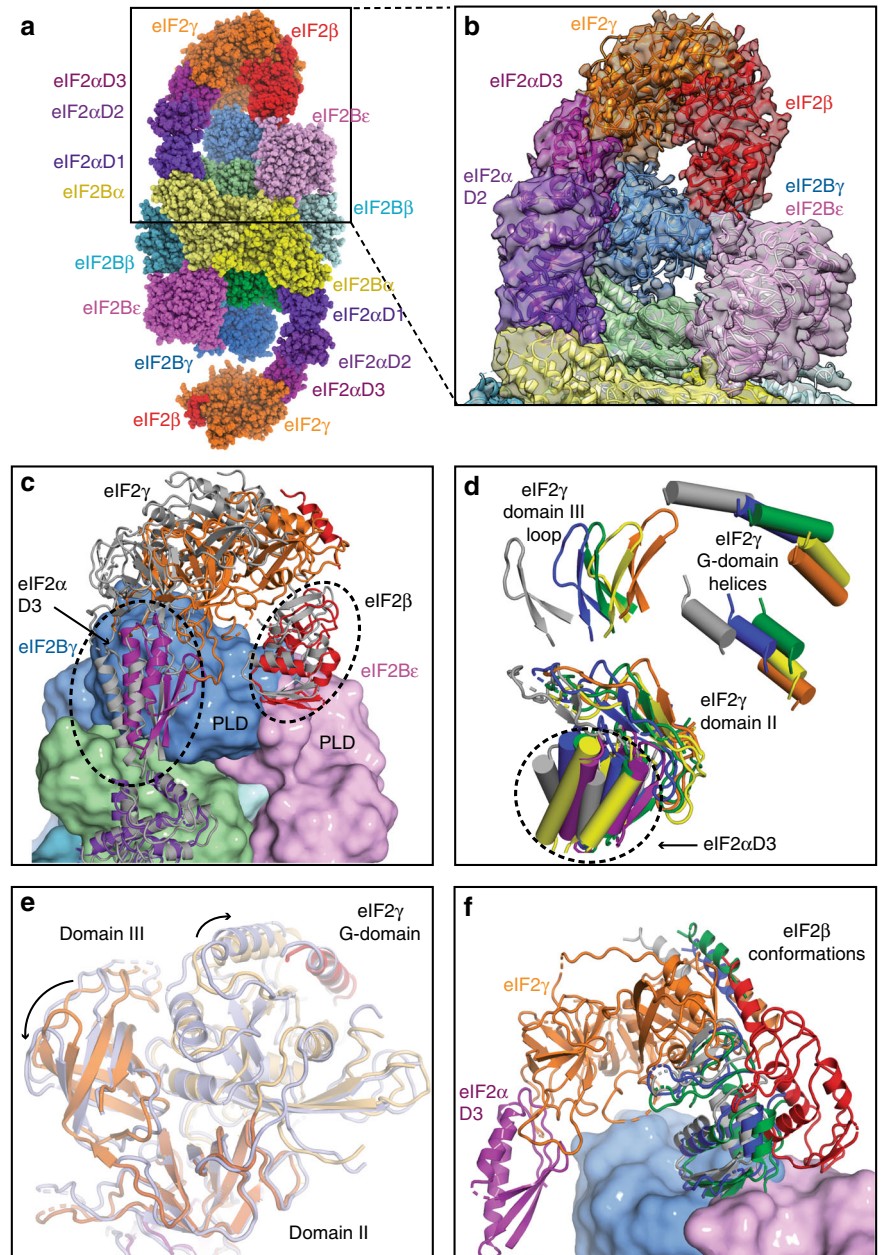

**Fig. 3** Contacts of eIF2 β and γ with the catalytic subunits of eIF2B. **a** eIF2B–eIF2(αP) complex model in spheres representation fitted in map A showing tilted conformation of eIF2γ, which is stabilised by its contact with eIF2Bγ and extended conformation of eIF2β contacting the interface area of the two γ and ε catalytic subunits of eIF2B. **b** Close-up view of the model fitting into the density of map A. **c** Modelled positions of eIF2 γ and β subunits after classification, showing extensive movements of these subunits around eIF2B ε and γ PLD domains. For clarity, only two eIF2 models are shown (corresponding to maps A—coloured and C—grey). **d** Modelled positions of eIF2γ subunit and eIF2α-D3 in all three maps (map 1—orange for eIF2γ and purple for eIF2α-D3, map A—blue, map B—green, map C—grey, map D—yellow). For clarity, only few elements in each of the eIF2γ domains are shown. **e** Superposition of domain II of eIF2γ in map A (coloured light blue) with that in map 1 (eIF2γ coloured orange) shows the rearrangement of three eIF2γ domains when it is in the tilted conformation. **f** Conformation of eIF2β fitted in map D (red) is different from the conformations found in three other maps (A - blue, B - green and C - grey). eIF2 γ and α shown are from map D

(corresponding to residues 97–101 and 136–139 in eIF2Bγ PLD). This conformation results in a slight rearrangement of the three domains in eIF2γ, compared to the TC structure (Fig. 3e)[7,9,41]. Also, the eIF2γ G-domain in this conformation is more disordered than in the TC, possibly reflecting a higher mobility of this domain in this particular conformation. Previously, a rearrangement of the three γ domains which depended on the nucleotide-binding state was reported in a crystallographic study in archaeal aIF2[42].

In all three maps, the density for eIF2β allowed modelling of the zinc-binding and central domains in the conformation similar to the one in the TC, but with the zinc-binding domain only partially covering the nucleotide-binding pocket and extended central domain approaching the binding interface between the eIF2B γ and ε PLD domains (Fig. 3f). One of these maps also contained extra density contacting the top of eIF2γ G-domain (Fig. 4a, b), large enough to accommodate eIF2B ε-cat HEAT domain in proximity to the N terminus of eIF2β, previously

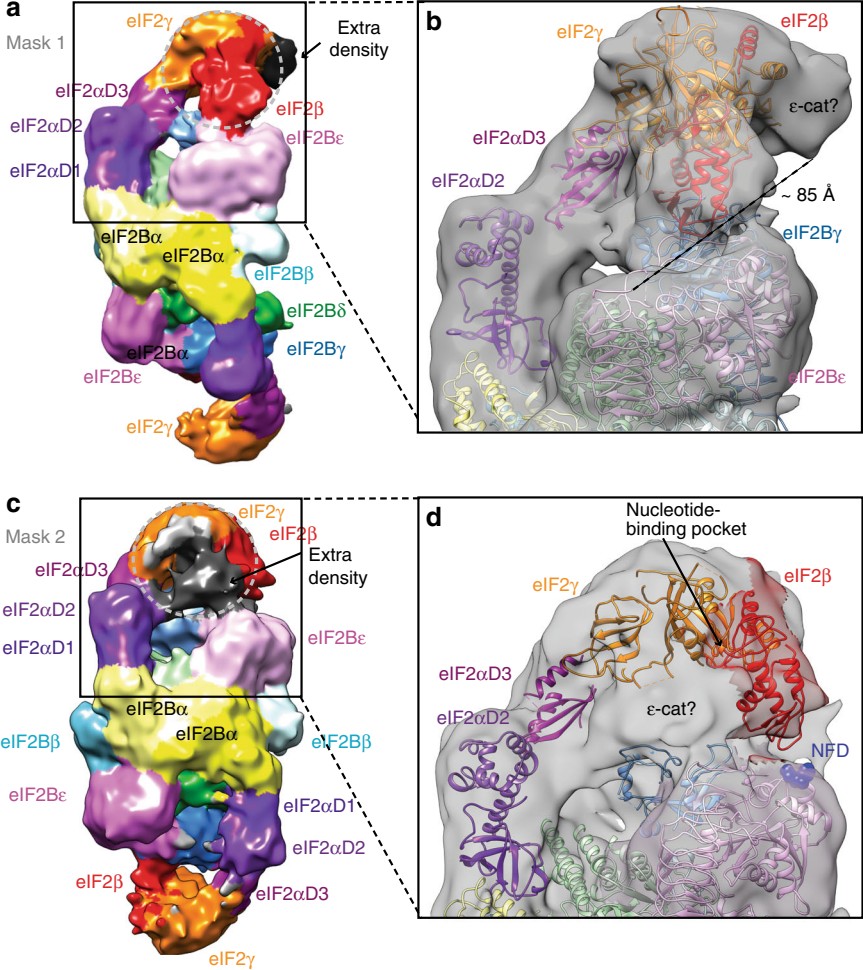

**Fig. 4** Extra densities in the maps after masked classification in proximity of eIF2γ could accommodate eIF2B ε-cat HEAT domain with eIF2. **a** Map B of eIF2B–eIF2(αP) complex obtained by masked classification around eIF2 γ and β showing extra density in contact with eIF2γ. **b** The extra density in map B could accommodate most of the ε-cat HEAT domain and would be in contact with eIF2γ domains III and G away from nucleotide-binding site. ~85 Å distance separates this extra density from the C terminus of the eIF2Bε and is just enough for the 73 residues linker (res. 472–544) to connect ε-cat with the rest of eIF2Bε. **c** Map D of eIF2B–eIF2(αP) complex obtained by masked classification around eIF2 γ and an extra density seen at a lower threshold (black) in proximity of nucleotide-binding pocket. **d** The size and shape of the extra density in map D could fully account for the whole ε-cat HEAT domain. Also, in this map, eIF2β approaches the NFD motif (blue spheres) in eIF2Bε

shown to interact with the ε-cat HEAT domain[43]. In this position, the ε-cat domain would not have access to the nucleotide-binding pocket on the eIF2γ-G-domain. However, we cannot exclude the possibility that eIF2B ε-cat could act allosterically by inducing rearrangement of the domains in eIF2γ, which we can see in the maps with the tilted conformations of eIF2γ, leading to nucleotide release. In this case the 73 residues linker (res. 472–544), connecting ε-cat with the rest of eIF2Bε, is just long enough to cover the distance of around 85 Å that separates this density from the C terminus of the modelled eIF2Bε (Fig. 4b).

A second masked classification yielded a map at only 10.4 Å resolution (Fig. 4c, d and Supplementary Fig. 1) but with a defined extra density, also of the size of eIF2B ε-cat domain, this time, on the other side of the eIF2γ-G-domain close enough to the nucleotide-binding region (Fig. 4d). Interestingly, this map also contained the density for eIF2β, not included in the mask. In this map eIF2β central domain now approaches the NF motif in eIF2Bε subunit, which is important for catalysis[34,36] (Fig. 4d), while zinc-binding domain, although not very well-defined, does not cover the nucleotide-binding pocket (red conformation of eIF2β in Fig. 3f).

## Discussion

The structure of eIF2B–eIF2(αP) complex, presented here, directly shows that two eIF2 molecules bind opposite sides of an eIF2B hetero-decamer. Although we do see particles of eIF2B alone, we do not observe particles corresponding to only one molecule of eIF2 bound to eIF2B in our datasets (Supplementary Fig. 1). Furthermore, each eIF2 molecule has bipartite interactions with eIF2B hetero-decamer—through eIF2α-D1 inserted in the pocket between eIF2B α and δ subunits and eIF2γ contacting catalytic eIF2B subunits. The interaction of eIF2α-D1 to the regulatory moiety of eIF2B is relatively well-defined in our structure, and likely makes the major contribution to the affinity between these two factors. In our complex eIF2α was phosphorylated in vitro at Ser51, which is known to result in an even more stable interaction with eIF2B[12,38]. The effect of Ser51 phosphorylation may be attributed to a combination of direct interactions with the residues in eIF2Bα and induced contact with eIF2Bδ. The large interaction area of eIF2α-D1 with eIF2B α and δ (844 and 374 Å$^2$, respectively), in between which eIF2α-D1 is sandwiched, implies that most of the contacts would be very similar even in the absence of phosphorylation. This conclusion is

also supported by cross-linking experiments[28] showing that the binding mode of eIF2α to the regulatory moiety of eIF2B is hardly affected by its phosphorylation status. However, the additional crosslinks which occurred in the absence of phosphorylation to Q91 and R84 of eIF2Bβ identified in the same study[28] (corresponding to E100 and A93 in our structure (Fig. 2b)) are far from the contact interface, suggesting that the binding of non-phosphorylated eIF2α may not be as stable.

Previously, eIF2(αP) has been shown to effectively sequester eIF2B[44,45], but also act as a competitive inhibitor of nucleotide exchange and prevent catalysis by non-productive interactions of eIF2(αP) with eIF2Bε-cat[21]. The local resolution in eIF2 γ, β and eIF2B ε-cat does not allow us to elucidate the details of nucleotide displacement. However, inhibition of the nucleotide exchange by eIF2α phosphorylation in the same molecule would not account on its own, for relatively small proportion of phosphorylated eIF2 (~30%) sufficient for inhibiting eIF2B activity[45], as the majority of non-phosphorylated eIF2 still would be available for productive nucleotide exchange even with limiting amounts of eIF2B in the cell. In contrast, the idea of sequestration of the much less abundant eIF2B when compared to eIF2 (ten times less[46]), seems the most important reason for translation inhibition by eIF2 phosphorylation, especially since binding of eIF2(αP) to the regulatory subunits of eIF2B is enhanced when compared to its unphosphorylated form and necessary for the inhibition of translation[12].

Recently, Jennings et al.[47] showed that nucleotides have a minor impact on the overall affinity of eIF2 to eIF2B using affinity pull-down, likely reflecting the fact that binding of eIF2 to the regulatory core of eIF2B through α-D1 makes the major contribution to the affinity and masked the interactions with the catalytic eIF2B subunits. Our reconstructions of the eIF2B–eIF2(αP) complex show high mobility and flexibility of eIF2 γ and β around catalytic portion of eIF2B, while maintaining the stronger contact through eIF2α-D1. The ratio of GTP to GDP (10:1) in the cell would be preferable for initial binding of GTP to eIF2 after GDP displacement by the catalytic portion of eIF2B as association rates of the nucleotides are comparable[48]. However, release of GTP by eIF2 is much faster than that of GDP[48] and therefore the equilibrium must be shifted by Met-tRNA$_i$$^{Met}$ binding to eIF2γ-GTP. The acceptor stem of Met-tRNA$_i$$^{Met}$ mainly contributes to the affinity of eIF2 binding in the TC[49,50], suggesting that this contact is driving formation of the TC and could occur while eIF2α-D1 is still being attached to regulatory portion of eIF2B. In fact a stable interaction of eIF2B in complex with GTP-eIF2 and Met-tRNA$_i$$^{Met}$ has been shown previously[44]. Superposition of eIF2 bound to eIF2B in our complex with the eIF2 structure in the TC[7] shows that this interaction is possible in the context of eIF2B–eIF2(αP) complex (Fig. 5a).

For the completion of the TC formation, a large conformational change in eIF2α is needed (Fig. 5b, c), as eIF2α-D1 must be extracted from eIF2B, as both Met-tRNA$_i$$^{Met}$ and eIF2Bδ share the same binding interface with the helix 58–63 in eIF2α. This shared binding interface creates direct competition between Met-tRNA$_i$$^{Met}$ and eIF2B for eIF2 binding. The competition between eIF2B and Met-tRNA$_i$$^{Met}$ for eIF2 binding has been recently shown experimentally[47]. Our structure suggests that Ser51-P directly interacts with residues in eIF2Bα, and that phosphorylation of eIF2α Ser51 slightly alters the conformation of the helix 58–63 in eIF2α, which may tip the balance towards eIF2B binding and prevent TC formation.

At the same time, the competition for eIF2 between eIF2B and Met-tRNA$_i$$^{Met}$ is also influenced by the competition for eIF2 between eIF2B ε-cat and eIF5-CTD[43,51,52], which share the same fold. Both eIF2B ε-cat and eIF5-CTD bind the eIF2γ-G-domain as well as the same region in eIF2β[43,51,52], the former displacing

the nucleotide and the latter protecting it from displacement[47,53,54]. While eIF2B was shown to disrupt TC[47], adding eIF5 or eIF5-CTD to the TC protected it from disruption, but not when eIF2α is phosphorylated. These data suggest that there is a fine balance between the catalytic and regulatory interactions of eIF2 and eIF2B, which are affected by other binding partners—eIF5 and Met-tRNA$_i$$^{Met}$. We propose, that it is not eIF2B that discriminates between the nucleotide states of eIF2, but rather subsequent interactions with Met-tRNA$_i$$^{Met}$ allow this discrimination in the cell. In fact the presence of Met-tRNA$_i$$^{Met}$ has been shown to stimulate the rate of GDP to GTP exchange by eIF2B[44,55].

Sequestering of eIF2B by phosphorylated eIF2, which is present in cell in ~10 times excess, has been suggested as a mechanism of ISR based on a number of biochemical studies[12,45] and generally is in agreement with the structure of eIF2B–eIF2(αP) complex that we have obtained. However, the sequestration does not necessarily have to be irreversible. A slow dissociation rate of eIF2 (αP) would prevent high turnover of eIF2B recycling and subsequent binding to non-phosphorylated eIF2. Therefore, the picture emerges that GEF and ISR function of eIF2B are structurally coupled and driven kinetically by the further formation of the TC (and eIF5 binding)—proceeding to initiation.

At the time of submission of our manuscript another three groups deposited manuscripts in bioRxiv with the structures of the yeast[56,57] and human[57,58] eIF2B–eIF2 complexes. In yeast both phosphorylated and non-phosphorylated eIF2α bound between eIF2B α and δ subunits with minor differences in the arginine-rich loop following Ser51[56,57]. Interestingly, the structures of human eIF2B–eIF2 complexes show different binding modes of eIF2 to eIF2B depending on the state of phosphorylation of eIF2. While human eIF2(αP) also binds between eIF2B α and δ subunits, non-phosphorylated human eIF2α binds to an alternative binding site—between β and δ subunits[57,58] with nucleotide exchange taking place on the other side of eIF2B hetero-decamer.

The sequences of eIF2α are very well conserved across species. To see the nature and extent of any differences in human and yeast eIF2B regulatory subunits, which constitute the binding sites for eIF2α, we aligned S. cerevisiae, S. pombe and human eIF2B sequences (Supplementary Fig. 6). The residues in eIF2B α and δ which constitute eIF2(αP) binding interface and residues in eIF2Bδ, which interact with eIF2α in either binding site, are well conserved (Supplementary Fig. 6a and b). However, the eIF2Bβ residues in a "tethering loop" (Y137-T148 in S. pombe and L117-K129 in S. cerevisiae) which binds eIF2Bα in the vicinity of eIF2α binding pocket between eIF2B α and δ subunits (Fig. 2b and Supplementary Fig. 6c) are truncated in human eIF2Bβ (Supplementary Fig. 6c). Mutations in the tether of eIF2Bβ I118T and S119P were shown to reduce the effect of phosphorylation in yeast[40]. Furthermore, residues in human eIF2Bβ (N132, E135) making direct contacts with residues in eIF2α in the alternative binding site[58] are not conserved in yeast, although they are strictly conserved at least in vertebrates (Supplementary Fig. 6c). The corresponding residues in S. pombe V153, Q156 and residue D160 in close proximity did not show any crosslink to eIF2α[28]. The majority of the high intensity crosslinks to non-phosphorylated eIF2α also mapped in the pocket between α and δ subunits, apart from two low intensity crosslinks to eIF2Bβ R84 and Q91[28] in the helix adjacent to the alternative binding site in human. In addition, in the same study[28] many crosslinks were found between eIF2γ and eIF2Bβ for both phosphorylated and non-phosphorylated eIF2. These crosslinks obtained in yeast factors are not consistent with the alternative eIF2 binding mode described in human[57,58]. Instead they are more consistent with both non-phosphorylated and phosphorylated eIF2 binding

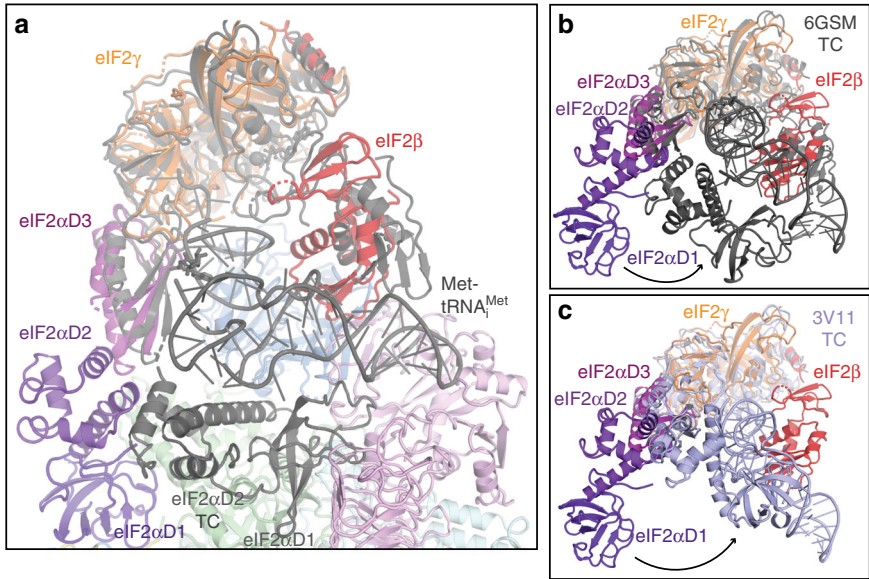

**Fig. 5** Superposition of the TC with eIF2B–eIF2(αP) complex based on eIF2γ. **a** Superposition of yeast TC (6GSM) in grey with the model of eIF2B–eIF2(αP) complex in map C showing that Met-tRNAᵢMet can bind without clash to eIF2γ and eIF2α- D3 while eIF2α-D1 is still attached to eIF2B. **b** Same superposition as in **a** in a different orientation shows the large conformational changes that eIF2α–D1 and D2 undergo when bound to eIF2B or Met-tRNAᵢMet both competing for eIF2α. **c** Same as in **b**, but superimposed with *S. solfataricus* TC (3V11, light blue)

between eIF2B α and δ subunits. However, crosslinks to eIF2γ identified in eIF2Bε[28] could be consistent with either binding mode. Therefore, we cannot entirely exclude the existence of the two eIF2α binding sites with different affinities in yeast eIF2B, although it is possible that the alternative binding site for eIF2 in eIF2B between β and δ subunits evolved later (in vertebrates), for example, allowing a more efficient nucleotide exchange on the other side of eIF2B hetero-decamer at the expense of the stability of the factor, which resulted in eIF2Bα being necessary to maintain the decameric structure of the eIF2B complex in human. The extra density we found in map B (Fig. 4b) would be in a similar location with respect to eIF2γ as eIF2B ε-cat HEAT domain in the structures obtained in[57,58], suggesting the possibility of nucleotide exchange on the opposite side of eIF2B hetero-decamer in yeast.

The recently isolated ISR inhibitor (ISRIB)[59] was used in one of the human structures[58] to stabilise the binding of non-phosphorylated eIF2 to eIF2B. ISRIB was shown to bind human eIF2B at the twofold symmetric interface "stapling" two βδ dimers of the regulatory core[26,27] and boost the "catalytic activity" of eIF2B in both phosphorylated and non-phosphorylated eIF2[26,59–61]. Its action was mostly attributed to the stabilisation of the eIF2B hetero-decamer in human[26,61], which is less stable than in yeast[62,63]. Comparison of ISRIB bound eIF2B with our eIF2 (αP) bound eIF2B structure (Supplementary Fig. 3) shows that ISRIB imposes a distinct symmetric eIF2B structure, which is incompatible with stable binding of two eIF2(αP) molecules in the pocket between α and δ subunits at the same time (Supplementary Fig. 3d, e) by precluding complete closure of eIF2Bδ helical bundle NTD around eIF2(αP)-D1. Therefore, ISRIB seems not only stabilise eIF2B hetero-decamer, but also impose a particular conformation of eIF2B regulatory core leading to a slight closure between β and δ NTDs comprising an alternative binding site for non-phosphorylated eIF2α in human and thereby selecting for binding of eIF2α over eIF2(αP).

## Methods

**Protein purification and complex assembly.** *Saccharomyces cerevisiae* eIF2 was purified from yeast strain GP3511 (*MATα leu2-3 leu2-112 ura3-52::HIS4-lacZ ino1 gcn2Δ pep4::LEU2 sui2Δ* pAV1089[*SUI2 SUI3 GCD11*-His6 2 μm *URA3*])[38]

as described previously[64]. Prior to assembly of the complex with eIF2B, purified eIF2 was phosphorylated in vitro by human PKR (Invitrogen)[65,66]. Phosphorylation of eIF2α Ser51(52) was confirmed by western blotting using antibodies specific against human eIF2α(P) (Invitrogen 44–728 G) (Supplementary Fig. 7b) and was measured by mass spectrometry to be 89.4% (Supplementary Fig. 7c).

*S. cerevisiae* eIF2B was over-expressed in yeast strain GP4109 (*MATα leu2-3 leu2-112 ura3-52 ino1 gcd6Δ gcn2Δ::hisG ura3-52::HIS4-lacZ* pAV1428[*GCD6 GCD1*-FLAG2-His6 *URA3* 2 μm] pAV1494[*GCN3 GCD2 GCD7 LEU2* 2 μm])[31]. After harvesting cells were suspended 1:1 (w:v) in PBS and cell suspension droplets were frozen in liquid nitrogen. Usually 50 g of cell "popcorn" was used for each purification of the eIF2B in complex with phosphorylated eIF2 (eIF2(αP)). After cell lysis Flag-tagged eIF2B complexes were immobilised on 300 μl of Anti-Flag M2 affinity gel (Sigma) and washed with a high-salt buffer (500 mM KCl)[67] followed by phosphorylation buffer (20 mM Tris (pH 7.5), 100 mM KCl, 10 mM MgCl₂, 5 mM β-glycerophosphate, 2 mM dithiothreitol, 10% glycerine, 0.1% NP-40, 200 μM ATP). The amount of eIF2B was estimated not to exceed 200 μg from 50 g of cell "popcorn" based on repeated purifications of eIF2B on its own. Therefore, in our phosphorylation reaction we used over ~2 times equimolar amount of eIF2 assuming two molecules of eIF2 can bind one eIF2B hetero-decamer. Phosphorylation reaction containing 2 mM GDP was added to immobilised Flag-tagged eIF2B and incubated at room temperature for 20 min. Beads were washed twice with the buffer—20 mM Hepes (pH 7.5), 100 mM KCl, 5 mM MgCl₂, 5 mM β-ME. eIF2B–eIF2(αP) complexes were eluted in 250 μl of the same buffer containing 100 μg/ml of 3XFlag-peptide (Sigma) and washed/concentrated in Amicon Ultra 50 K MWCO concentrators 5 times in the buffer without 3XFlag-peptide. Protein concentration was measured by nanodrop and Bradford reaction, which gave concentration values within 10% difference, usually in the range of 1 to 2 μg/μl or 1.17 to 2.35 μM (assuming two molecules of eIF2 bind eIF2B hetero-decamer) (Supplementary Fig. 7a).

Immediately before applying to cryo grids, the sample was diluted five to ten times to ~200 nM with the same buffer containing glutaraldehyde to make the final concentration of the glutaraldehyde 0.1% (concentration of the glutaraldehyde in the buffer added to the sample did not exceed 0.125%).

**Electron microscopy.** Three μl of the eIF2B–eIF2(αP) complex were applied to glow-discharged gold UltrAuFoil R 1.2/1.3 or R 2/2 grids at 4 °C and 100% ambient humidity. After 30 s incubation, the grids were blotted for 4–5 s and vitrified in liquid ethane using a Vitrobot Mk3 (Thermo Fisher Scientific).

Automated data acquisition was done using the EPU software (Thermo Fisher Scientific) on a Titan Krios microscope (Thermo Fisher Scientific) operated at 300 kV under low-dose conditions in linear (dataset I, 45 e⁻/Å²) or counting mode (dataset II, 21 e⁻/Å²) using a defocus range of 1.5–4.5 μm. In linear mode, images of 1.1 s/exposure and 34 movie frames were recorded (Supplementary Fig. 8a), whereas in counting mode, we saved 75 fractions over a 60 s exposure, using in both cases a Falcon III direct electron detector (Thermo Fisher Scientific) at a calibrated magnification of 104,478 (yielding a pixel size of 1.34 Å). Micrographs that showed noticeable signs of astigmatism or drift were discarded.

**Analysis and structure determination**. The movie frames were aligned with MotionCor2[68] for whole-image motion correction. Contrast transfer function parameters for the micrographs were estimated using Gctf[69]. Particles were picked using Relion[70]. References for template-based particle picking[71] were obtained from 2D class averages that were calculated from particles semi-automatically picked with EMAN2[72] from a subset of the micrographs. For dataset 2, the references for template-based particle picking were obtained from 2D class averages of the eIF2B–eIF2 complex map at 5.7 Å (see below). 2D class averaging (Supplementary Fig. 8c), 3D classification and refinements were done using RELION-2[70]. Both movie processing[73] in RELION-2 and particle "polishing" were performed for all selected particles for 3D refinement. Resolutions reported here (Supplementary Fig. 8b) are based on the gold-standard FSC = 0.143 criterion[74]. All maps were further processed for the modulation transfer function of the detector, and sharpened[75]. Local resolution was estimated using ResMap[76].

For the dataset I, 3282 images were recorded from two independent data acquisition sessions, and 459,480 particles were selected after two-dimensional classification. An initial 3D reconstruction was made from all selected particles after 2D class averaging using the *Schizosaccharomyces pombe* eIF2B crystal structure (PDB: 5B04) low-pass filtered to 60 Å as an initial model, and using internal C2 symmetry. Next, two consecutive 3D classification into 15 and 6 classes, respectively, this time without using the eIF2B internal symmetry, with a 7.5 degrees angular sampling interval and no local searches was performed to remove bad particles or empty eIF2B particles from the data and to get an initial understanding of the conformational heterogeneity of eIF2 in the complex. After the second round of 3D classification, 239,695 particles were selected (52% of the total) and refined to 5.7 Å resolution.

The map did not yield a high overall resolution, partly due to limited distribution of orientation (Supplementary Fig. 8d); therefore, we collected an additional dataset using a different grid from the same batch at the same magnification and using the same detector but in counting instead of linear mode. For this dataset (dataset II), 1241 images were recorded, and 173,740 particles were selected after two-dimensional classification. After obtaining an initial three-dimensional refined model, and two consecutive rounds of 3D classification the classes containing the eIF2B–eIF2(αP) complex were selected (131,663 particles, 75% of the total) and after movie processing, refined using C2 internal symmetry to much higher resolution than for the dataset I (map 1, 4.2 Å).

The particles from both datasets were then combined and a masked 3D classification using masks around two eIF2γ molecules in the complex was carried out to remove particles with low occupancy for these factors, as a result of which 183,468 particles were selected and refined to 4.3 Å (map 2). The overall resolution of this map was slightly lower than that of map 1, but the occupancy and local resolution for eIF2γ and eIF2α-D3 was better.

The preliminary 3D rounds of classification showed that eIF2γ, eIF2α-D3, and densities possibly belonging to eIF2β and the HEAT domain of eIF2Bε adopt many different conformations. So we carried out 3D classifications with subtraction of the residual signal[77] by creating two different masks—one around the density attributed to eIF2α-D3, eIF2γ and eIF2β in all possible conformations observed in the preliminary 3D classification rounds, and another around a density observed at low threshold in close proximity to the eIF2γ G-domain. We applied these masks for each of the two molecules of eIF2 in each eIF2B–eIF2(αP) complex. We isolated four distinct and well-defined maps by 'focused' 3D classifications, as follows:

(a) Map A, showing higher occupancy for eIF2β and a tilted conformation of eIF2γ [119,037 particles, 4.6 Å];

(b) Map B, similar to map A but with slightly different conformations of eIF2β and eIF2γ. It also shows an extra density in contact with the G-domain and domain III of eIF2γ [12,575 particles, 9.4 Å];

(c) Map C, showing the most extreme tilted conformation towards eIF2Bγ for eIF2γ, and where eIF2β is also observed [23,909 particles, 10.1 Å];

(d) Map D, showing additional density in contact with eIF2γ, whose size and shape suggested that it could correspond to eIF2B ε-cat HEAT domain [23,909 particles, 10.4 Å].

**Model building and refinement**. In all six maps the conformations of all eIF2B subunits and domains D1 and D2 of eIF2α are nearly identical. Thus, modelling of all these elements was first done in the higher resolution maps (4.2 and 4.3 Å; maps 1 and 2), and then this model was used as a reference for model building in EM maps with lower resolution (maps A to D). In this procedure, the crystal structure model of eIF2B from *S. pombe* (PDB: 5B04) was placed into density by rigid-body fitting using Chimera[78]. Then each subunit of eIF2B was independently fitted by rigid-body refinement, first in Chimera and then in Coot[79]. Also in Coot, the sequence was converted to that of *S. cerevisiae* proteins, followed by rigid-body fitting of different subdomains within each eIF2B subunit. Further modelling was also done in Coot, paying special attention to the region of eIF2B in contact with eIF2.

eIF2 was taken from PDB: 6FYX. eIF2α-D1/eIF2α-D2 and eIF2α-D3/eIF2γ/eIF2β N-terminal helix were fitted as separate rigid bodies into its corresponding densities, using Chimera and Coot. Then, each of these domains but the eIF2β n-terminal helix was independently fitted, and further modelling was also done in Coot.

## Table 1 CryoEM data collection, refinement and validation statistics

| | Map 1 (EMDB-4543) (PDB 6QG0) | Map A (EMDB-4545) (PDB 6QG2) |
|---|---|---|
| **Data collection and processing** | | |
| Magnification | 104,478 | 104,478 |
| Voltage (kV) | 300 | 300 |
| Electron exposure (e$^-$/Å$^2$) | 45 | 21 |
| Defocus range (μm) | 1.5–4.5 | 1.5–4.5 |
| Pixel size (Å) | 1.34 | 1.34 |
| Symmetry imposed | C2 | C1 |
| Initial particle images (no.) | 173,740 | 633,220 |
| Final particle images (no.) | 131,663 | 119,037 |
| Map resolution (Å) | 4.2 | 4.6 |
| 　FSC threshold | 0.143 | 0.143 |
| Map resolution range (Å) | — | — |
| **Refinement** | | |
| Initial model used (PDB code) | 5B04 | 5B04 |
| Model resolution (Å) | 4.2 | 4.6 |
| 　FSC | 0.45 | 0.42 |
| Model resolution range (Å) | — | — |
| Map sharpening *B* factor (Å$^2$) | −119 | −100 |
| Model composition | | |
| 　Non-hydrogen atoms | 36,980 | 38,676 |
| 　Protein residues | 4,742 | 4,961 |
| 　Ligands | — | — |
| *B* factors (Å$^2$) | 356 | — |
| 　Protein | 356 | — |
| 　Ligand | — | — |
| R.m.s. deviations | | |
| 　Bond lengths (Å) | 0.008 | 0.008 |
| 　Bond angles (°) | 1.24 | 1.258 |
| Validation | | |
| 　MolProbity score | 2.48 (99th) | 2.50 (98th) |
| 　Clashscore | 5.3 (100th) | 3.68 (100th) |
| 　Poor rotamers (%) | 5.9 | 21.2 |
| Ramachandran plot | | |
| 　Favoured (%) | 87.6 | 83.3 |
| 　Allowed (%) | 10 | 13.2 |
| 　Disallowed (%) | 2.4 | 3.5 |

Model refinement in the highest resolution maps was carried out in Refmac v5.8 optimised for electron microscopy[80], using external restraints generated by ProSMART[80]. The average Fourier Shell Coefficient (FSC) was monitored during refinement. The final model was validated using MolProbity[81]. Cross-validation against overfitting (Supplementary Fig. 8e) was done as previously described[80,82]. Refinement statistics for the last refinements, done in Map 1, are given in Table 1.

These refined models were used as initial models for maps A-D, and then each subunit of the model was rigid-body fitted, without observing almost any appreciable change, except for the eIF2α-D3/eIF2γ/eIF2β N-terminal helix sub-module in one of the two eIF2 molecules. After the fitting of this eIF2α-D3/eIF2γ/eIF2β N-terminal helix sub-module in each of these maps, an extra density belonging to the whole eIF2β subunit was observed and we consequently docked into it the subunit β from PDB: 6FYX. In map D, although there is density for most of eIF2β, it was not possible to do an appropriate rigid-body docking without any major clashes and we decided not to include eIF2β in the final model. We also did not include eIF2B ε-cat HEAT domain in any of the models in maps B or D due to the poor local resolution.

All figures were generated using PyMOL, Coot or Chimera. Analysis of particle orientation distribution was done with CryoEF[83].

**Multiple sequence alignment**. Multiple sequence alignment of eIF2B sequences was done using Clustal Omega[84].

**Reporting summary**. Further information on research design is available in the Nature Research Reporting Summary linked to this article.

## Data availability

The data that support the findings of this study are available from the corresponding author upon request. Six maps have been deposited in the EMDB with accession codes

EMD-4543, EMD-4544, EMD-4545, EMD-4546, EMD-4547, EMD-4548, for Map 1, Map 2, Map A, Map B, Map C and Map D, respectively. Six atomic coordinate models have been deposited in the PDB with accession codes 6QG0, 6QG1, 6QG2, 6QG3, 6QG5 and 6QG6 for Maps 1, 2, Map A, Map B, Map C and Map D, respectively.

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

## Acknowledgements

We thank G. Pavitt for providing *Saccharomyces cerevisiae* strain GP4109 for over-expressing yeast eIF2B. We thank Mark Skehel and Sarah Maslen for MS analysis. We are grateful to G. Cannone and G. McMullan for technical support with cryoEM, T. Darling and J. Grimmett for help with computing. This study was supported by the MRC-LMB EM Facility. This work was supported by grants from the Medical Research Council (MC_U105184332) and the Wellcome Trust (WT096570) to V.R. and by a grant BFU2017-85814-P from the Spanish government to J.L.L.

## Author contributions

Y.G. conceived the study, purified the protein complex and prepared the samples. J.L.L. and Y.G. performed electron cryo-microscopy data collection. J.L.L. performed processing, model building and analysis. Y.G. and J.L.L. wrote the first draft of the manuscript. V.R. helped edit and revise the manuscript.

## Additional information

**Competing interests:** The authors declare no competing interests.

**Journal Peer Review Information:** *Nature Communications* thanks Madhusudan Dey, and the other, anonymous, reviewer(s) for their contribution to the peer review of this work. Peer reviewer reports are available.

