## [Peer Review File · Nature Communications]

Reviewers' comments:

Reviewer #1 (Remarks to the Author):

The manuscript presents the first cryo-EM structure of eIF2B in complex with two molecules of phosphorylated eIF2. The description of the structure allows to link and explain a wealth of biochemical and genetic data published over the last years about the regulation of eIF2 nucleotide exchange, a key step involved in stress regulation and cell fitness. Nevertheless I have the feeling that the cryo-EM structure alone presented here might not be enough to fully support all the results and conclusions drawn in this study, mainly because of the limited resolution of the map (4 Å at its very best).

1) The first part of the results presenting the overall structure is sometimes unclear and hard to follow:

- first paragraph: how did applying a two-fold C2 symmetry help for dealing with conformational heterogeneity?
- first paragraph: "Additionally, we obtained four density maps..." In addition of what exactly?
- second paragraph: "As judged by the local resolution (Fig. S2), the most stable..." How is local resolution linked with stability?
- second paragraph: "This interaction is possibly further enhanced by phosphorylation of eIF2 α in our complex" Can the authors support this assertion with a reference or complementary experiments?
- second paragraph: "This contact has lower resolution,..." I don't think a "contact" has a resolution, I would rather talk about a "region", "area" or something similar.
- third paragraph: "In two of these maps additional low-resolution density could be attributed to the ϵ -cat heat domain..." How?? The local resolution seems to be 10 Å or worth, no secondary structure features are visible, making any identification or rigid body fitting highly hazardous! Do the authors have complementary biochemical and/or biophysical evidences to support identification of the ϵ -cat heat domain in their maps?

2) The interaction of the phosphorylated Ser51 with eIF2B is obviously a key point of the manuscript but the conclusions need to be supported more clearly by the results and the figures:

- Is Ser51 100% phosphorylated?
- Fig. S4a and S4b are not clear enough to judge the quality of the densities in this region; to talk about precise residues at such a resolution (~4 Å), one needs to be convinced.

3) Discussion:

- "... the binding of eIF2 to eIF2B is cooperative." The authors needs at least a reference or complementary experiments to support this assertion.
- A figure presenting their model would be useful.

4) The authors should include the FSC curve of the model vs the map, as well as the FSC curves of the cross-validation against overfitting.

Minor comments:

- The authors are using sometimes one, sometimes two numbers after the decimal point for their resolutions: they should use one.
- FEI company doesn't exist any more; it is now Thermo Fisher Scientific.
- in the "Methods" section, maps 1 and 2 are called I and II.
- in the "Methods" section, PDB: 5B04 is derived from a crystal structure, not cryo-EM.
- in the "Methods" section, if the buffer used to dilute the sample before freezing contains 0.125% glutaraldehyde, then the final concentration of glutaraldehyde should be 0.01% and not 0.1%.

- Figure S2: the same resolution range should be used for all the maps.
- Figure S5: the micrograph needs a scale bar; the 2D classes could be better presented, with less classes and only a sampling of representative views.
- An angular distribution plot for all the maps could be also interesting.

Reviewer #2 (Remarks to the Author):

The decameric eIF2B is the guanine nucleotide exchange factor (GEF) for eIF2, a trimeric GTPase that brings the initiator Met-tRNA_i to the ribosome. eIF2B is one of the main targets of regulation of protein synthesis. The substrate eIF2 is phosphorylated by several stress-induced kinases, in what is collectively known as the Integrated Stress Response (ISR). Phosphorylated eIF2 (eIF2(a-P)) acts as a competitive inhibitor of eIF2B. Dysregulated SR is implicated in a number of neurodegenerative disorders, including Alzheimer's Disease. The mechanisms of eIF2B action and regulation are of great scientific and medical importance and are currently the subject of high interest and intensive research by multiple labs in the field. In fact, currently three other manuscripts reporting the Cryo-EM structures of eIF2B:eIF2 (enzyme:substrate) and/or eIF2B:eIF2(a-P) (enzyme:inhibitor) complexes have been deposited pre-publication in BioRxiv.

This manuscript reports the Cryo-EM structure of the eIF2B:eIF2(a-P) complex. A mechanism for the inhibition of eIF2B activity by phosphorylation of eIF2 is proposed. The authors also propose a mechanism for the catalytic activity of eIF2B. However, that model may be misguided, because it is based on the assumption that the eIF2B:eIF2 complex is similar to the eIF2B:eIF2(a-P) complex. While one other manuscript, from the Pavitt lab, reports a low resolution structure of eIF2B:eIF2 that seems similar to the eIF2B:eIF2(a-P) structure, high-resolution eIF2B:eIF2 structures solved by the Ito lab and the Walter lab, and deposited in BioRxiv, show drastically different structures. The work from the Ito lab is especially of note because it reports the structures of both the eIF2B:eIF2 (enzyme:substrate) and/or eIF2B:eIF2(a-P) (enzyme:inhibitor) complexes, and they are extremely different: essentially, mirror images along the two-fold symmetry of eIF2B.

To be clear, ordinarily, the assumption that the active and inhibited complexes of eIF2B:eIF2 are similar would be a very solid one (and everyone in the field has shared it for decades), if it weren't for the two newly-solved structures, which show otherwise, and which explain plethora of experimental data.

Specific Comments

p. 5, line 101: Note that the helix bundle domains of eIF2B α , β and δ are the NTDs, not the CTDs. The hexamer assembly is through the CTDs, while the helical NTDs protrude and form the eIF2 α -binding pockets. Correct here and throughout the manuscript.

p. 5, lines 122-123. The crosslinks to eIF2B β from Kashiwagi et al., 2016 that are inconsistent with the structure reported here, and are also weaker with phosphorylated eIF2 α , are consistent with the structures of the complex of eIF2B with unphosphorylated eIF2, deposited to BioRxiv, by the Walter and Ito labs. The authors should add a comment to that effect.

p. 8, lines 189 to the end of the paragraph, as well as throughout the manuscript. See also the previous comment. In the eIF2B:eIF2 structures from the Ito and Walter labs, the overall arrangement is completely different, including the orientation of the catalytic domain with respect to the rest of eIF2B. Unless the authors have reason to believe that these two structures of eIF2B:eIF2 (enzyme:substrate) are incorrect, they should not use the eIF2B:eIF2(a-P) structure

(enzyme:inhibitor) and the position of the catalytic domain in it to interpret the catalytic mechanism. Instead, if the position of the catalytic domain is different, that would suggest a mechanism of inhibition by eIF2 α phosphorylation.

p. 8, line 189. "ε-cat heat" should be "eIF2Bε-cat HEAT".

p. 11, line 261. In citing Jennings et al. the authors should state that Jennings et al. reported that nucleotides have a minor impact on the affinity of eIF2 for eIF2B (using affinity pull-down, hardly a standard quantitative assay), and should also point out that the report was at odds with earlier reports that the affinity of apo-eIF2 is higher (see e.g. Goss et al., 1984, Panniers et al., 1988), and that the affinity of GDP and GTP for eIF2 is lower when eIF2 is bound to eIF2B (Panniers et al., 1988). Based on thermodynamic coupling, if GDP has lower affinity for eIF2B:eIF2 than for eIF2, then eIF2B has lower affinity for eIF2-GDP than for apo-eIF2. Higher affinity of eIF2B for the reaction intermediate, apo-eIF2, is also a requirement for eIF2B to promote GDP dissociation.

p. 11, lines 267-269. The authors state that "In the cell the probability of GTP binding by eIF2 after GDP displacement by the catalytic portion of eIF2B is much higher than that of GDP due to an approximately 10 times higher GTP concentration." This statement needs to be corrected. The ratio of GTP and GDP bound to eIF2B:eIF2 depends not only on the ratio between the concentrations of free GTP and GDP, but also in their K_d's for eIF2B:eIF2. It has been reported long ago that the relative K_d's of GTP and GDP for eIF2B:eIF2 are ~1:10, which with the GTP concentration being 10x higher yields ~1:1 ratio of eIF2B:eIF2:GTP to eIF2B:eIF2:GDP, which is still at least ten-fold more favorable than that for free eIF2. The idea that Met-tRNAⁱ plays a role in shifting the equilibrium toward the GTP-bound form, thus becomes even more relevant.

Reviewer #3 (Remarks to the Author):

During translation, an important step is formation of a ternary complex (TC) consisting of eIF2, initiator methionyl tRNA (Met-tRNAⁱMet) and GTP (eIF2-Met-tRNAⁱMet-GTP). TC delivers Met-tRNAⁱMet to the 40S ribosomal subunit, and then eIF2 dissociates as an eIF2-GDP form. The eIF2-GDP is recycled to eIF2-GTP by a guanine nucleotide exchange factor eIF2B, which interacts again with the Met-tRNAⁱMet to form a new TC for the next round of translation.

Both eIF2 and eIF2B are multi-subunit proteins. eIF2 consists of α , β and γ subunit, whereas eIF2B consists of α , β , γ , δ , and ϵ subunits. Under conditions of cellular stress, α -subunit of eIF2 is phosphorylated on the residue Ser51. The phosphorylated form of eIF2 then binds tightly to eIF2B, thus inhibiting its GTP exchange activity. A large number of genetic and biochemical data support this concept; however, the structural information was not available on the binding mode of eIF2B and eIF2. Here, authors resolved the cryo-EM structure of eIF2B bound to the phosphorylated form of eIF2 [eIF2B-eIF2(α P)], purified from the budding yeast *Saccharomyces cerevisiae*.

The eIF2B-eIF2 α P structure reveals that each of the five subunits of eIF2B (α , β , γ , δ , and ϵ) assembled into a decamer of dimers, consistent with the published structure of the fission yeast or human eIF2B. This structure also reveals that the phosphorylated form of eIF2 α binds mainly to the α -subunit of eIF2B. Overall, this complex structure provides direct evidence on how eIF2 binds to eIF2B, which significantly contribute to our understanding the translational control by eIF2 α phosphorylation.

Authors have provided a detailed discussion in the context of the published genetic and biochemical data. But their unique observations have not been supported by genetic and/or biochemical data.

Answers to Reviewer's comments:

Our responses are in BLUE

Reviewer #1 (Remarks to the Author): The manuscript presents the first cryo-EM structure of eIF2B in complex with two molecules of phosphorylated eIF2. The description of the structure allows to link and explain a wealth of biochemical and genetic data published over the last years about the regulation of eIF2 nucleotide exchange, a key step involved in stress regulation and cell fitness. Nevertheless I have the feeling that the cryo-EM structure alone presented here might not be enough to fully support all the results and conclusions drawn in this study, mainly because of the limited resolution of the map (4.00Å at its very best):

1) The first part of the results presenting the overall structure is sometimes unclear and hard to follow:

- first paragraph: how did applying a two-fold C2 symmetry help for dealing with conformational heterogeneity?

Response: In order to maximize the resolution in the less heterogeneous parts of the model we applied a two-fold C2 symmetry refinement.

- first paragraph: "Additionally, we obtained four density maps..." In addition of what exactly?

Response: In addition to the two maps obtained by applying a two-fold C2 symmetry EM data processing, we obtained four density maps without applying any symmetry.

We have now modified the text to make it clearer for the reader as follows: "This structure was obtained by applying a two-fold C2 symmetry during EM data processing (maps 1 and 2, Fig. S1) resulting in maximum resolution for the most homogeneous parts of the model but an averaged position for the eIF2 molecules. To account for the different conformations of eIF2 in the complex, we also carried out EM data classifications without applying any internal symmetry and we obtained another four density maps (maps A to D in Fig. S1), however at the expense of the overall resolution."

second paragraph: "As judged by the local resolution (Fig. S2), the most stable..." How is local resolution linked with stability?

Response: If positions of molecules are slightly different due to local flexibility or motion, this will result in the loss of detail in density map. We wanted to emphasize the fact that in this region, where the resolution is relatively high ~ 4.2 Å, there is less local flexibility and motion compared to the other parts of the molecules in the complex and therefore this contact is more rigid or strong. We have now replaced this sentence with "As judged by the relatively high local resolution (Fig. S2), which reflects low local flexibility and mobility, the strongest contact consists of eIF2 \$\alpha\$ domain D1 inserted between the N-terminal helix bundle domains of \$\alpha\$ and \$\delta\$ regulatory subunits of eIF2B."

second paragraph: "This interaction is possibly further enhanced by phosphorylation of eIF2 α in our complex" Can the authors support this assertion with a reference or complementary experiments?

We added two references: Krishnamoorthy, T., Pavitt, G. D., Zhang, F., Dever, T. E. & Hinnebusch, A. G. Tight binding of the phosphorylated alpha subunit of initiation

factor 2 (eIF2 α) to the regulatory subunits of guanine nucleotide exchange factor eIF2B is required for inhibition of translation initiation. *Mol Cell Biol* **21**, 5018-5030 (2001).

Pavitt, G. D., Ramaiah, K. V., Kimball, S. R. & Hinnebusch, A. G. eIF2 independently binds two distinct eIF2B subcomplexes that catalyze and regulate guanine-nucleotide exchange. *Genes Dev* **12**, 514-526 (1998).

- second paragraph: "This contact has lower resolution,..." I don't think a "contact" has a resolution, I would rather talk about a "region", "area" or something similar. As suggested, we have replaced "contact" by "area of contact"

- third paragraph: "In two of these maps additional low-resolution density could be attributed to the ϵ -cat heat domain..." How?? The local resolution seems to be 10 Å or worth, no secondary structure features are visible, making any identification or rigid body fitting highly hazardous! Do the authors have complementary biochemical and/or biophysical evidences to support identification of the ϵ -cat heat domain in their maps?

Response: We have accounted for all the densities in our reconstruction corresponding to domains in eIF2B-eIF2(α P) apart from eIF2B ϵ -cat HEAT domain which is present in our purified complex. Although the resolution is poor, size wise the density is big enough to accommodate ϵ -cat HEAT domain and this extra density is in contact with eIF2 γ where you would expect ϵ -cat domain to interact. In one of the maps (map B) the density is located where the heat domain is shown to bind relative to eIF2 γ (manuscripts in BioRxiv deposited few days after we submitted our manuscript, which we now cite). We did not include this domain in our models and only referred to the extra densities as possible locations of the ϵ -cat domain. However figure 4b and 4d in the manuscript could have been misleading and therefore we removed the model of heat domain from the maps in these figures and only marked the area with labels. We removed also removed referral to the ϵ -cat HEAT domain in the first part of the results section.

2) The interaction of the phosphorylated Ser51 with eIF2B is obviously a key point of the manuscript but the conclusions need to be supported more clearly by the results and the figures:

- Is Ser51 100% phosphorylated?

Response: Phosphorylation of eIF2 α Ser51(52) was determined previously by mass spectrometry {Gordiyenko et al., 2014} to be 98 % using the same experimental conditions as described here. We now included western blotting using antibodies specific against human eIF2 α (P) (Invitrogen 44-728G) and mass spectrometry analysis showing 89.4% of S51 phosphorylation in our eIF2 sample (Fig. S7 b and c).

- Fig. S4a and S4b are not clear enough to judge the quality of the densities in this region; to talk about precise residues at such a resolution (~4 Å), one needs to be convinced.

Response: We changed figure S4a and b to show the density around the phosphorylated S51 with more clarity. We also show now an alternative modelling of this region according to ({Kashiwagi K, 2018} and {Adomavicius T, 2018}) results, which do not fit as well in our density.

3) Discussion:

- "... the binding of eIF2 to eIF2B is cooperative." The authors need at least a reference or complementary experiments to support this assertion.

Response: Given that we do not have complementary experiments to support this assertion, we have removed that sentence.

- A figure presenting their model would be useful.

Response: We would prefer not to provide a model since it would be too speculative because it is not clear what the structure of eIF2B-eIF2 in the active conformation in yeast looks like.

4) The authors should include the FSC curve of the model vs the map, as well as the FSC curves of the cross-validation against overfitting.

Response: We have provided it now (Fig. S8b)

Minor comments:

- The authors are using sometimes one, sometimes two numbers after the decimal point for their resolutions: they should use one. Response: We have corrected this.

- FEI company doesn't exist any more; it is now Thermo Fisher Scientific. Response: We have corrected this.

- in the "Methods" section, maps 1 and 2 are called I and II. Response: We have corrected this.

- in the "Methods" section, PDB: 5B04 is derived from a crystal structure, not cryo-EM. Response: We have corrected this.

- in the "Methods" section, if the buffer used to dilute the sample before freezing contains 0.125% glutaraldehyde, then the final concentration of glutaraldehyde should be 0.01% and not 0.1%. Response: Final glutaraldehyde concentration was 0.1% and we now changed the wording in the text to clarify this point.

- Figure S2: the same resolution range should be used for all the maps. Response: We would prefer to keep the figure as it is, since the overall resolution between the two maps represented differs too much to have the same resolution range.

- Figure S5: the micrograph needs a scale bar; the 2D classes could be better presented, with less classes and only a sampling of representative views. Response: We have included a scale bar in the micrograph and we now show only the representative views (now Fig. S8).

- An angular distribution plot for all the maps could be also interesting. Response: This would require a full figure and we would prefer to not replace any of the current figures in the paper.

Reviewer #2 (Remarks to the Author):

The decameric eIF2B is the guanine nucleotide exchange factor (GEF) for eIF2, a trimeric GTPase that brings the initiator Met-tRNA_i to the ribosome. eIF2B is one of the main targets of regulation of protein synthesis. The substrate eIF2 is phosphorylated by several stress-induced kinases, in what is collectively known as the Integrated Stress Response (ISR). Phosphorylated eIF2 (eIF2(α -P)) acts as a competitive inhibitor of eIF2B. Dysregulated SR is implicated in a number of neurodegenerative disorders, including Alzheimer's Disease. The mechanisms of eIF2B action and regulation are of great scientific and medical importance and are currently the subject of high interest and intensive research by multiple labs in the field. In fact, currently three other manuscripts reporting the Cryo-EM structures of

eIF2B:eIF2 (enzyme:substrate) and/or eIF2B:eIF2(α -P) (enzyme:inhibitor) complexes have been deposited pre-publication in BioRxiv.

This manuscript reports the Cryo-EM structure of the eIF2B:eIF2(α -P) complex. A mechanism for the inhibition of eIF2B activity by phosphorylation of eIF2 is proposed. The authors also propose a mechanism for the catalytic activity of eIF2B. However, that model may be misguided, because it is based on the assumption that the eIF2B:eIF2 complex is similar to the eIF2B:eIF2(α -P) complex. While one other manuscript, from the Pavitt lab, reports a low resolution structure of eIF2B:eIF2 that seems similar to the eIF2B:eIF2(α -P) structure, high-resolution eIF2B:eIF2 structures solved by the Ito lab and the Walter lab, and deposited in BioRxiv, show drastically different structures. The work from the Ito lab is especially of note because it reports the structures of both the eIF2B:eIF2 (enzyme:substrate) and/or eIF2B:eIF2(α -P) (enzyme:inhibitor) complexes, and they are extremely different: essentially, mirror images along the two-fold symmetry of eIF2B.

To be clear, ordinarily, the assumption that the active and inhibited complexes of eIF2B:eIF2 are similar would be a very solid one (and everyone in the field has shared it for decades), if it weren't for the two newly-solved structures, which show otherwise, and which explain plethora of experimental data.

Response: We thank the reviewer for the thorough analysis of our paper and identifying a few mistakes in the manuscript as well as the comments concerning the mechanism of the nucleotide exchange in eIF2 by eIF2B and also rising a question of the lack of similarity between the eIF2B-eIF2 and eIF2B-eIF2(α P) in the light of the manuscripts deposited in BioRxiv by the Ito's lab (21/12/2018) and the Frost's lab (22/12/2018) at the same time as our paper was submitted (21/12/2018).

The Cryo-EM structures presented by these groups clearly show different binding modes of eIF2 and eIF2(α P) to eIF2B in human factors (called "productive" and "non-productive" binding, respectively), however we would like to draw the reviewer's attention that Ito's lab crystal structures of yeast factors – *S. cerevisiae* eIF2 α and eIF2 α P bound to *S. pombe* eIF2B show that both phosphorylated and non phosphorylated eIF2 α subunits bound in the same place – between eIF2B α and δ subunits, as in our structure, with minor differences in the Ser51-flanking loop and the short α -helix after, which interacts with eIF2B δ . That is in agreement with the conclusion we made based on our structure of eIF2B-eIF2(α P) in yeast that both structures should be similar. Ito and colleagues attributed this result to the fact that these structures were obtained with eIF2 α subunits only. However in the fourth manuscript deposited in BioRxiv by the Pavitt's group (20/12/2018) both eIF2B-eIF2 and eIF2B-eIF2(α P) cryo-EM structures (from *S. cerevisiae*) are identical to our complex, with eIF2 α or eIF2 α P sandwiched in between eIF2B α and δ subunits, again with minor differences.

Although a few (two) specific cross-links of non- phosphorylated eIF2 α to eIF2B β identified by Kashiwagi et.al in yeast seem to be in agreement with the alternative binding site for eIF2 α found in human eIF2B-eIF2 complex, there are more cross-links (also found in phosphorylated eIF2 α) which mapped in the pocket between α and δ subunits. Also bands for these specific cross-links are of much lower intensity compared to those in the pocket between α and δ subunits in a non-phosphorylated eIF2. Furthermore cross-links to eIF2 γ identified in the catalytic eIF2B γ and ϵ subunits to both phosphorylated and non-phosphorylated eIF2 cover a

substantial area of eIF2B γ and differ in two cross-links in eIF2B ϵ absent in phosphorylated eIF2. Substantial interaction of yeast eIF2 γ and eIF2B γ in both phosphorylated and non-phosphorylated eIF2 is consistent with the data obtained by us and Pavitt's group, while in human eIF2B-eIF2 structures there is hardly any contact between eIF2 γ and eIF2B γ when non-phosphorylated eIF2 α binds to alternative binding site. To investigate if there are any differences in human and yeast eIF2B regulatory subunits, which constitute binding sites for eIF2 α , we aligned *S.cerevisiae*, *S. pombe* and human eIF2B sequences. In fact, residues of eIF2B β interacting with eIF2 α described in Frost's and Ito's papers are not conserved in yeast eIF2B β (new Fig. S6) and none of them (mutated to pBP α) crosslinked to eIF2 α in Kashiwagi et.al. Furthermore, human eIF2B β is missing part of the loop containing a tether to eIF2B α . This tether region interacts with eIF2 α in yeast.

We then aligned a few more eIF2B β sequences from different species (new Fig. S6c) and found that alternative binding site between eIF2B β and δ subunits is conserved at least in vertebrates. This suggests that a second eIF2 α binding site (the "productive binding site") evolved in human eIF2B (not present in yeast), possibly allowing a more efficient nucleotide exchange on the other side of eIF2B hetero-decamer. However we cannot entirely exclude the existence of two eIF2 α binding sites with different affinities in yeast eIF2B at this time. We believe this subject needs more investigation, which is outside the scope of this manuscript. However we have made changes to discussion in our manuscript to acknowledge the structures deposited by other groups and to interpret our results in the light of the other structures.

Specific Comments

p. 5, line 101: Note that the helix bundle domains of eIF2B α , β and δ are the NTDs, not the CTDs. The hexamer assembly is through the CTDs, while the helical NTDs protrude and form the eIF2 α -binding pockets. Correct here and throughout the manuscript.

Response: We have corrected domains assignment in regulatory eIF2B subunits.

p. 5, lines 122-123. The crosslinks to eIF2B β from Kashiwagi et al., 2016 that are inconsistent with the structure reported here, and are also weaker with phosphorylated eIF2 α , are consistent with the structures of the complex of eIF2B with unphosphorylated eIF2, deposited to BioRxiv, by the Walter and Ito labs. The authors should add a comment to that effect.

Response: While the crosslinks in common for the unphosphorylated/phosphorylated eIF2 are consistent with the conformation shown here, the crosslinks to eIF2B β that are weaker with phosphorylated eIF2 α , are close but not in the interface in these structures of the complex of eIF2B with unphosphorylated eIF2 (see figure S8 in Ito's paper). In fact the residues of eIF2B β in direct contact with eIF2 α (N132, E135 and E139) are hardly conserved in yeast (correspond to V153, Q156 and D160, respectively) and did not shown any crosslink in Kashiwagi et al., 2016.

p. 8, lines 189 to the end of the paragraph, as well as throughout the manuscript. See also the previous comment. In the eIF2B:eIF2 structures from the Ito and Walter labs, the overall arrangement is completely different, including the orientation of the catalytic domain with respect to the rest of eIF2B. Unless the authors have reason to

believe that these two structures of eIF2B:eIF2 (enzyme:substrate) are incorrect, they should not use the eIF2B:eIF2(a-P) structure (enzyme:inhibitor) and the position of the catalytic domain in it to interpret the catalytic mechanism. Instead, if the position of the catalytic domain is different, that would suggest a mechanism of inhibition by eIF2 α phosphorylation.

Response: We would like to stress that we only make minimal referral to the catalytic mechanism of nucleotide exchange per se in the absence of the high-resolution data in this region. However we feel that we should not omit the extra densities in some of the classes that we obtained, although at different positions with respect to eIF2 γ . In fact one of the extra densities we see in map B is in a similar location with respect to eIF2 γ as in the structures obtained by the Frost and Ito labs, but on the other side of eIF2B hetero-decamer.

p. 8, line 189. “ ϵ -cat heat” should be “eIF2B ϵ -cat HEAT”. We have corrected this.

p. 11, line 261. In citing Jennings et al. the authors should state that Jennings et al. reported that nucleotides have a minor impact on the affinity of eIF2 for eIF2B (using affinity pull-down, hardly a standard quantitative assay), and should also point out that the report was at odds with earlier reports that the affinity of apo-eIF2 is higher (see e.g. Goss et al., 1984, Panniers et al., 1988), and that the affinity of GDP and GTP for eIF2 is lower when eIF2 is bound to eIF2B (Panniers et al., 1988). Based on thermodynamic coupling, if GDP has lower affinity for eIF2B:eIF2 than for eIF2, then eIF2B has lower affinity for eIF2-GDP than for apo-eIF2. Higher affinity of eIF2B for the reaction intermediate, apo-eIF2, is also a requirement for eIF2B to promote GDP dissociation.

p. 11, lines 267-269. The authors state that “In the cell the probability of GTP binding by eIF2 after GDP displacement by the catalytic portion of eIF2B is much higher than that of GDP due to an approximately 10 times higher GTP concentration”;

This statement needs to be corrected. The ratio of GTP and GDP bound to eIF2B:eIF2 depends not only on the ratio between the concentrations of free GTP and GDP, but also in their K_d 's for eIF2B:eIF2. It has been reported long ago that the relative K_d 's of GTP and GDP for eIF2B:eIF2 are ~1:10, which with the GTP concentration being 10x higher yields ~1:1 ratio of eIF2B:eIF2:GTP to eIF2B:eIF2:GDP, which is still at least ten-fold more favorable than that for free eIF2. The idea that Met-tRNA_i plays a role in shifting the equilibrium toward the GTP-bound form, thus becomes even more relevant.

Response: We agree with the reviewer that the ratio of GTP and GDP bound to eIF2B-eIF2 complex at equilibrium depends not only on the ratio of GTP and GDP, but also on their K_d s (GTP:GDP ~ 1:10). In the paper the reviewer refers to, Panniers et al showed, that the difference in K_d s is due to much faster GTP release with the rate constants for binding being approximately equal. Therefore Met-tRNA_i binding to eIF2-GTP would be required to shift the equilibrium to prevent fast GTP dissociation. We made changes in the text to clarify this point according to the reviewer's comments.

We also thank the reviewer for pointing out the importance of Met-tRNA_i in shifting the equilibrium towards eIF2-GTP. We believe that the question of affinity between eIF2 and eIF2B and the influence of nucleotides on this affinity is quite complex in the view of the newly solved structures, which show bipartite interactions

of eIF2 with eIF2B. We agree that thermodynamic coupling apply for one of these interactions, namely interaction of eIF2 γ subunit which contains nucleotide binding site, with the catalytic portion of eIF2B (ϵ and γ subunits), however another interaction which is formed by eIF2 α with the regulatory eIF2B subunits does not seem to depend on nucleotide state of eIF2. The affinity measured by Jennings et. al in pull-down experiments most likely reflects combined interactions of eIF2 and eIF2B masking the catalytic interaction. We have now stated in the text that Jennings et.al measured affinity by pull-down experiments.

The bipartite mode of interaction between eIF2 and eIF2B would also influence the interpretation of the results obtained by Goss et al and Panniers et al. Goss et al measured affinity between eIF2 and eIF2B in the presence and absence of GDP by fluorescence anisotropy of dansyl chloride labelled eIF2. The change in fluorescence polarisation would occur even in the case when only catalytic interaction was broken in accordance with the structural evidence that when bound to eIF2B through eIF2 α -D1, the rest of eIF2 is quite mobile. We agree that eIF2B would lower the affinity of GDP or GTP to eIF2 γ due to catalytic interaction, however it would not influence the interaction of eIF2 α with the regulatory eIF2B subunits and therefore we propose that Met-tRNA^{iMet} is required to extract eIF2 α from eIF2B.

Reviewer #3 (Remarks to the Author):

During translation, an important step is formation of a ternary complex (TC) consisting of eIF2, initiator methionyl tRNA (Met-tRNA^{iMet}) and GTP (eIF2-Met-tRNA^{iMet}-GTP). TC delivers Met-tRNA^{iMet} to the 40S ribosomal subunit, and then eIF2 dissociates as an eIF2-GDP form. The eIF2-GDP is recycled to eIF2-GTP by a guanine nucleotide exchange factor eIF2B, which interacts again with the Met-tRNA^{iMet} to form a new TC for the next round of translation.

Both eIF2 and eIF2B are multi-subunit proteins. eIF2 consists of α , β , and γ subunit, whereas eIF2B consists of α , β , γ , δ and ϵ subunits. Under conditions of cellular stress, α -subunit of eIF2 is phosphorylated on the residue Ser51. The phosphorylated form of eIF2 then binds tightly to eIF2B, thus inhibiting its GTP exchange activity. A large number of genetic and biochemical data support this concept; however, the structural information was not available on the binding mode of eIF2B and eIF2. Here, authors resolved the cryo-EM structure of eIF2B bound to the phosphorylated form of eIF2 [eIF2B-eIF2(α P)], purified from the budding yeast *Saccharomyces cerevisiae*.

The eIF2B-eIF2 α P structure reveals that each of the five subunits of eIF2B (α , β , γ , δ and ϵ) assembled into a decamer of dimers, consistent with the published structure of the fission yeast or human eIF2B. This structure also reveals that the phosphorylated form of eIF2 α binds mainly to the α -subunit of eIF2B. Overall, this complex structure provides direct evidence on how eIF2 binds to eIF2B, which significantly contribute to our understanding the translational control by eIF2 α phosphorylation.

Authors have provided a detailed discussion in the context of the published genetic and biochemical data. But their unique observations have not been supported by genetic and/or biochemical data.

Response: We thank the reviewer for pointing this. Over the last three decades a plethora of genetic and biochemical data were published about the regulation of eIF2 nucleotide exchange by eIF2B and interactions between these two factors, however

precise structural information was still missing. The vast majority of the published genetic and biochemical data validates the structure/s described in this paper. Moreover structural information obtained here provides new insight to the previously published biochemical data by showing two spatially separated contacts of one eIF2 molecule with regulatory and catalytic moieties of eIF2B and structural evidence of direct Met-tRNA^{iMet} competition with eIF2B for eIF2 α . In the revised version of this manuscript we have now included western blotting using antibodies specific against human eIF2 α (P) and mass spectrometry analysis showing 89.4% of S51 phosphorylation in our eIF2 sample (see figure S7).

REVIEWERS' COMMENTS:

Reviewer #1 (Remarks to the Author):

I acknowledge that the authors took into consideration most of my remarks.

Without questioning the quality of the data, I am still concerned about how the cryo-EM results are described and presented: since it is the only result of this study, it is important that everything is easy to understand.

- Two different datasets were acquired for this study, using two very different imaging conditions (Falcon 3 in linear and counting mode). Did the authors use the same grid for both sessions? Or two grids from the same freezing batch?
- After merging the two datasets, what is the proportion of dataset 1 versus dataset 2 in the different maps (2, A, B, C and D).

- Is there an explanation why there is 52% of eIF2-eIF2B particles in the first dataset and 75% in the second?

- The first paragraph of the result describing the overall structure of eIF2B-eIF2 starts with map 1 (at 4.2 Å resolution), but then the rest of the manuscript (including Figure 1) only presents map 2. Furthermore, it seems that the models for both maps are the same. It would be good for the author to clarify a bit more this first part of the results.

- It appears that 10% of the eIF2B pool is not phosphorylated; is there a chance that one of the small subclasses (C or D) corresponds to a non-phosphorylated state?

- It seems clear that the second dataset is better in terms of image quality: did the authors try to perform a similar analysis using this dataset only?

- I would still insist to add the orientation distribution plots for the maps in one of the supplementary figures; it is a very valuable result shown in the big majority of all recent cryo-EM studies.

More generally the authors incorporated very well their results in the wealth of already published genetic/biochemical data, but because of the limited amount of results presented and because of the rather limited resolution of the structure, most of the proposed assertions are only putative. I am therefore very guarded about accepting this manuscript in its current form for publication in Nature Communications.

Reviewer #2 (Remarks to the Author):

The authors have addressed the main points from the initial review. I have no major concerns at this time.

Minor comments:

1. Page 10, first paragraph. Please, discuss also whether cross-links in Kashiwagi et al., 2016, between eIF2 γ and eIF2 ϵ are consistent with your structure. It seems that only the cross-links between eIF2 γ and eIF2 β are consistent, whereas to also account for the eIF2 γ -eIF2 ϵ cross-links, one may need a mix of the two alternative eIF2B:eIF2 complexes, as suggested in Kashiwagi et al., 2018.

2. Page 11, lines 263-266. Please, note that, while not immediately obvious for a bipartite interaction, thermodynamics still requires that the enzyme have higher affinity for the reaction intermediate, in order to speed up the reaction. If the affinity of the anchoring interaction remains the same, higher affinity for the reaction intermediate (in this case eIF2B:apo-eIF2) would lead to higher overall affinity. The physical explanation is that each interaction increases the effective concentration of the two partners with respect to each other at the other interface, and thus the binding on-rate. Likewise and irrespective of whether the interaction is bipartite, if eIF2B lowers the eIF2 affinity for GDP, GDP must lower the affinity of eIF2 for eIF2B. Otherwise, eIF2B would change the equilibrium between free substrate (eIF2-GDP), reaction intermediate (apo-eIF2), and product (eIF2-GTP), which no enzyme can do.

Reviewer #3 (Remarks to the Author):

This manuscript and the accompanied paper (Pavitt group) present the first cryo-EM structure of eIF2B in complex with phosphorylated eIF2, which significantly contribute to our understanding the translational control by eIF2 α phosphorylation. As I said previously, this manuscript alone lacks sufficient experimental evidence to support their structure.

Reviewer #1 (Remarks to the Author): I acknowledge that the authors took into consideration most of my remarks. Without questioning the quality of the data, I am still concerned about how the cryo-EM results are described and presented: since it is the only result of this study, it is important that everything is easy to understand.

- Two different datasets were acquired for this study, using two very different imaging conditions (Falcon 3 in linear and counting mode). Did the authors use the same grid for both sessions? Or two grids from the same freezing batch?

Response: different grids from the same batch were used for acquiring dataset 1 and dataset 2. We have now included the following sentence in the methods section: “The map did not yield a high overall resolution, partly due to limited orientation distribution of orientation (Supplementary Figure 8d); therefore we collected an additional dataset using a different grid from the same batch at the same magnification and using the same detector but in counting instead of linear mode”.

- After merging the two datasets, what is the proportion of dataset 1 versus dataset 2 in the different maps (2, A, B, C and D).

Response: For maps B, C and D the proportion of dataset 1 vs dataset 2 is about 2:1, which corresponds quite well with the proportion just after merging all particles of eIF2B-eIF2 complex from datasets 1 and 2; however in the higher resolution maps, the proportion of particles from dataset 2 (counting mode dataset) is higher. In the map 2 the percentage of particles is around 50% for each dataset, and for the map A the percentage of particles is about 60% and 40% for the dataset 1 and dataset 2, respectively. Therefore it seems that particles from dataset 2 contribute the most to improve the resolution in the maps 2 and A. However we would like to note that for map 2, if we take only particles from dataset 2 (around 90.000 particles), the resolution is worse than 4.4Å and therefore it was still beneficial in terms of resolution to merge both datasets instead of taking particles from dataset 2 only.

- Is there an explanation why there is 52% of eIF2-eIF2B particles in the first dataset and 75% in the second?

Response: there are several contributing factors to the percentage of particles included in the final maps. 1) The higher number of particles containing eIF2 (75%) in the second dataset is due to better quality of the data and therefore better alignment of the particles and fewer particles being discarded. In fact in dataset 1, 52% does not mean all the rest of particles do not contain eIF2, and for example some discarded particles in the second round of classification still contained some density for eIF2 α -D1 but these poorly aligned particles were classified out during classification process. 2) Automated particle picking in the dataset 2 was done with 2D class averages from particles from the 5.7 Å eIF2B-eIF2 map from dataset 1. However references for template-based particle picking in dataset 1 were obtained from 2D class averages from particles from a small subset of the micrographs and therefore the quality of the template was not as good as for dataset 2. The consequence of this is that more “junk particles” were picked, and not all were discarded in the 2D classification steps and therefore these “junk particles” had to be discarded in the first round of 3D classification. This is reflected in the quality of 2D class averages from datasets 1 (left panel, see figure below) and 2 (right panel, see figure below)

3) In addition the fact that we used 2D class averages from particles from a eIF2B-eIF2 complex for dataset 2 autopicking probably biased the percentages to have more particles of eIF2B-eIF2 complex instead of a mixture of empty eIF2B particles and eIF2B-eIF2 particles.

We added a sentence in the methods section detailing how the particles were picked for the second dataset. “For data set 2, the references for template-based particle picking were obtained from 2D class averages of the eIF2B-eIF2 complex map at 5.7 Å from data set 1(see below)”.

- The first paragraph of the result describing the overall structure of eIF2B-eIF2 starts with map 1 (at 4.2 Å resolution), but then the rest of the manuscript (including Figure 1) only presents map 2. Furthermore, it seems that the models for both maps are the same. It would be good for the author to clarify a bit more this first part of the results.

Response: Model for both maps 1 and 2 is essentially the same. The best overall resolution (4.2 Å) for the eIF2B-eIF2 α (P) complex was achieved after processing only dataset 2 acquired in counting mode. However due to high conformational heterogeneity and mobility of eIF2 molecules on the periphery of the complex, the number of particles in a particular conformation on the periphery is less in counting mode affecting local peripheral resolution. Combining particles from both datasets increased the number of particles in a particular conformation on the periphery of the complex. After focused classification on the eIF2 γ of the combined dataset, local resolution for eIF2 γ and eIF2 α -D3 improved (Map 2), although the overall resolution is slightly less 4.3 Å. Map 2 contains better density for eIF2 γ , eIF2 α -D3 and part of eIF2 β . Given that the resolution of maps 1 and 2 are similar and Map 2 provides more complete information about eIF2 we decided to use map 2 in Figure 1 and not map 1. The density maps provided in figures S4 and S5 are from Map 1, as this map has higher resolution in the core of the complex. In short, we used the best data we had that fit the purpose.

We have re-written the first paragraph of Results section to clarify how the cryoEM datasets were collected and how we obtained the different maps.

- It appears that 10% of the eIF2B pool is not phosphorylated; is there a chance that one of the small subclasses (C or D) corresponds to a non-phosphorylated state?

Response: We thank the reviewer for pointing out a possibility we did not consider previously. However we think it is unlikely that particles contributing to Map C correspond to the unphosphorylated form of eIF2 only since it contains more particles

than 10% of the total (23,909 out of 183,468); it would only be possible if there is an enrichment of particles corresponding to a complex with unphosphorylated eIF2 on the grid, which is very unlikely as we should expect the opposite since a complex with an unphosphorylated eIF2 is less stable to that with its phosphorylated form. In any case we cannot entirely exclude the possibility of some unphosphorylated eIF2 molecules present in this class taking into account that we had 90% phosphorylation efficiency. The same explanation is valid for map B, which contains even more particles, (32,759 particles). For class D this explanation is not entirely valid since it has only 12,575 particles (6.9% of the total). Nevertheless we would like to note that there are not global changes in the core of the eIF2B-eIF2 α (P) complex in none of these classes, in particular the region of binding eIF2 α (D1) in the pocket between α and δ subunits of eIF2B which presumably would allow a particular preferred conformation of the periphery of the complex in the case if eIF2 is not phosphorylated if that is the reviewer eluding to.

- It seems clear that the second dataset is better in term of images quality: did the authors try to perform a similar analysis using this dataset only?

Response: We agree with the reviewer that dataset 2 is better in terms of image quality (less noise), and therefore even with less number of particles (173,740) compared to the dataset 1 acquired in linear mode the overall resolution of eIF2B-eIF2 α (P) complex is better. We have processed each dataset independently and only for map 1 was worth it to have only the particles from data set 2. Even for map 2 that was obtained using a C2-fold symmetry, the merged data yields a better resolution (see response above). eIF2B-eIF2 α (P) complex has high conformational heterogeneity and mobility of eIF2 molecules on the periphery of the complex. Therefore combining particles from both datasets increased the number of particles in a particular conformation on the periphery of the complex and allowed to perform focused classification in this region. Focused classification, taking into account the particles from dataset 2 only, did not result in better resolution of the obtained classes most likely due to not enough particles in a particular conformation in the periphery.

- I would still insist to add the orientation distribution plots for the maps in one of the supplementary figures; it is a very valuable result shown in the big majority of all recent cryo-EM studies.

We added orientation distribution plot of the linear mode dataset 1 in Supplemental Figure 8d. We have also included an analysis of orientation distribution efficiency done with the program CryoEF and included a sentence in Methods section “The map did not yield a high overall resolution, partly due to limited orientation distribution of orientation (Supplementary Figure 8d); therefore we collected an additional dataset using a different grid from the same batch at the same magnification and using the same detector but in counting instead of linear mode ”

Reviewer #2 (Remarks to the Author): The authors have addressed the main points from the initial review. I have no major concerns at this time.

Minor comments:

1. Page 10, first paragraph. Please, discuss also whether cross-links in Kashiwagi et al., 2016, between eIF2 γ and eIF2 β are consistent with your structure.

It seems that only the cross-links between eIF2 γ and eIF2 β are consistent, whereas to also account for the eIF2 γ -eIF2 ϵ cross-links, one may need a mix of the two alternative eIF2B:eIF2 complexes, as suggested in Kashiwagi et al., 2018.

Response: we added referral to eIF2 ϵ cross-links on page 14 - "However crosslinks to eIF2 γ identified in eIF2B ϵ ²⁸ could be consistent with either binding mode" in Discussion section rather than in results section on page 10 as this information fitted better in discussion.

2. Page 11, lines 263-266. Please, note that, while not immediately obvious for a bipartite interaction, thermodynamics still requires that the enzyme have higher affinity for the reaction intermediate, in order to speed up the reaction. If the affinity of the anchoring interaction remains the same, higher affinity for the reaction intermediate (in this case eIF2B:apo-eIF2) would lead to higher overall affinity. The physical explanation is that each interaction increases the effective concentration of the two partners with respect to each other at the other interface, and thus the binding on-rate. Likewise and irrespective of whether the interaction is bipartite, if eIF2B lowers the eIF2 affinity for GDP, GDP must lower the affinity of eIF2 for eIF2B. Otherwise, eIF2B would change the equilibrium between free substrate (eIF2-GDP), reaction intermediate (apo-eIF2), and product (eIF2-GTP), which no enzyme can do.

Response: we have replaced **not** with **masked** in the sentence "Recently Jennings et al showed that nucleotides have a minor impact on the overall affinity of eIF2 to eIF2B⁴⁷ using affinity pull-down, likely reflecting the fact that binding of eIF2 to the regulatory core of eIF2B through α -D1 makes the major contribution to the affinity and **masked** the interactions with the catalytic eIF2B subunits".

Reviewer #3 (Remarks to the Author): This manuscript and the accompanied paper (Pavitt group) present the first cryo-EM structure of eIF2B in complex with phosphorylated eIF2, which significantly contribute to our understanding the translational control by eIF2 α ; phosphorylation. As I said previously, this manuscript alone lacks sufficient experimental evidence to support their structure.

Response: in terms of biochemistry the interactions between the two factors eIF2 and eIF2B were studied for over three decades, however until now there was no structure of the complex available to elucidate the interactions between these two factors. We feel that literature (references) provides a wealth of information to support the structure of the eIF2B-eIF2 α (P) complex that we obtained. Our structure is also in agreement with the structures of the same/similar complexes obtained by other groups (Ref. 56 to 58).